# Out-of-Distribution Optimality of Invariant Risk Minimization

**Shoji Toyota**                                                                 *shoji@ism.ac.jp*
*The Institute of Statistical Mathematics*

**Kenji Fukumizu**                                                               *fukumizu@ism.ac.jp*
*The Institute of Statistical Mathematics*

**Reviewed on OpenReview:** *https://openreview.net/forum?id=pWsfWDnJDa*

## Abstract

Deep Neural Networks often inherit spurious correlations embedded in training data and hence may fail to generalize to unseen domains, which have different distributions from the domain to provide training data. Arjovsky et al. (2019) introduced the concept *out-of-distribution (o.o.d.) risk*, which is the maximum risk among all domains, and formulated the issue caused by spurious correlations as a minimization problem of the o.o.d. risk. Invariant Risk Minimization (IRM) is considered to be a promising approach to minimize the o.o.d. risk: IRM estimates a minimum of the o.o.d. risk by solving a bi-level optimization problem. While IRM has attracted considerable attention with empirical success, it comes with few theoretical guarantees. Especially, a solid theoretical guarantee that the bi-level optimization problem gives the minimum of the o.o.d. risk has not yet been established. Aiming at providing a theoretical justification for IRM, this paper rigorously proves that a solution to the bi-level optimization problem minimizes the o.o.d. risk under certain conditions. The result also provides sufficient conditions on distributions providing training data and on a dimension of a feature space for the bi-leveled optimization problem to minimize the o.o.d. risk.

## 1 Introduction

Training data used in supervised learning may contain features that are spuriously correlated to the response variables of data. Deep Neural Networks (DNNs) often learn such spurious correlations embedded in the data and hence may fail to predict desirable response variables of test data generated by a distribution that is different from the one to provide training data. To list a few examples, in a classification of animal images, models obtained by conventional procedures tend to misclassify cows on sandy beaches because most training pictures are captured in green pastures and DNNs inherit context information in training (Beery et al., 2018; Shane, 2018). Another example is learning from medical data. Systems trained with data collected in one hospital do not generalize well to other hospitals; DNNs unintentionally extract environmental factors specific to a particular hospital in training (AlBadawy et al., 2018; Perone et al., 2019; Heaven, 2020).

Arjovsky et al. (2019) introduced the concept *out-of-distribution (o.o.d.) risk* to formulate the issue caused by spurious correlations. Let $\mathcal{X}$ and $\mathcal{Y}$ be measurable spaces of explanatory and response variables respectively. Let $\mathcal{E}$ be a set with each element $e \in \mathcal{E}$ called the *domain* (or environment) $e$. Assume that for a given domain $e \in \mathcal{E}$, there corresponds a corresponding random variable $(X^e, Y^e)$ that takes values in $\mathcal{X} \times \mathcal{Y}$ with its probability law $P_{X^e,Y^e}$. Assume we are given training datasets $\mathcal{D}^e := \{(x_i^e, y_i^e)\}_{i=1}^{n^e} \sim P_{X^e,Y^e}$ i.i.d. from multiple domains $\mathcal{E}_{tr} \subset \mathcal{E}$. For a given predictor $f : \mathcal{X} \to \mathcal{Y}$,

$$\mathcal{R}^e(f) := \int l(f(x), y) dP_{X^e, Y^e}$$

denotes the risk of $f$ on domain $e$. The o.o.d. risk of the predictor $f$ is as follows:

$$\mathcal{R}^{o.o.d.}(f) := \max_{e \in \mathcal{E}} \mathcal{R}^e(f), \tag{1}$$

which is the worst-case risk over $\mathcal{E}$ including unseen domain $\mathcal{E} - \mathcal{E}_{tr}$. Arjovsky et al. (2019) formulated the problem caused by spurious correlations as a minimization problem of the o.o.d. risk (1):

$$\min_{f \in \mathcal{F}} \mathcal{R}^{o.o.d.}(f), \tag{2}$$

where $\mathcal{F}$ is the set of all measurable functions $f : \mathcal{X} \to \mathcal{Y}$.

It is difficult to directly solve the o.o.d. risk minimization (2) since we can not evaluate the maximum of risks among all domains $\mathcal{E}$, including unseen domains $\mathcal{E} - \mathcal{E}_{tr}$, only by data from training domains $\mathcal{E}_{tr} \subset \mathcal{E}$. Invariant Risk Minimization (IRM) is a rapidly developing approach to the challenging o.o.d. risk minimization (Arjovsky et al., 2019). Its proposed predictor $f := w \circ \Phi$ is composed of two maps: a feature map $\Phi : \mathcal{X} \to \mathcal{H}$, which is called an *invariance*, and a predictor $w : \mathcal{H} \to \mathcal{Y}$ which estimates the response variable of feature $\Phi(x)$. Here, for a given feature space $\mathcal{H}$, we call a measurable function $\Phi : \mathcal{X} \to \mathcal{H}$ an invariance when it holds that $P_{Y^{e_1}|\Phi(X^{e_1})} = P_{Y^{e_2}|\Phi(X^{e_2})}$ for any $e_1, e_2 \in \mathcal{E}$[1]. Arjovsky et al. (2019) estimated the two maps by solving the bi-leveled optimization problem

$$\min_{\Phi \in \mathcal{I}_{tr}, w \in \mathcal{W}} \sum_{e \in \mathcal{E}_{tr}} \mathcal{R}^e(w \circ \Phi), \tag{3}$$

where $\mathcal{W}$ is a model of predictors $w : \mathcal{H} \to \mathcal{Y}$ and $\mathcal{I}_{tr}$ is the set of invariances captured by training domains $\mathcal{E}_{tr}$:

$$\mathcal{I}_{tr} := \left\{ \Phi : \mathcal{X} \to \mathcal{H} \,|\, P_{Y^{e_1}|\Phi(X^{e_1})} = P_{Y^{e_2}|\Phi(X^{e_2})} \text{ for any } e_1, e_2 \in \mathcal{E}_{tr} \right\}. \tag{4}$$

Influenced by the seminal study, several alternative bi-leveled optimization problems have been proposed (Ahuja et al., 2020; Chang et al., 2020; Ahuja et al., 2021a;b; Lin et al., 2022a; Zhou et al., 2022; Liu et al., 2021a;b; Lu et al., 2022; Koyama & Yamaguchi, 2021; Parascandolo et al., 2022; Krueger et al., 2021; Toyota & Fukumizu, 2022; Lin et al., 2022b; Huh & Baidya, 2022; Rame et al., 2022; Pogodin et al., 2023; Chen et al., 2023; Tan et al., 2023). For example, Ahuja et al. (2020) proposed a novel bi-leveled optimization problem leveraging the principles of game theory. The recently proposed Maximal Invariant Predictor (Koyama & Yamaguchi, 2021) employed a new bi-leveled problem grounded in the concept of information theory.

While IRM is widely recognized as a promising approach for the o.o.d. risk minimization (2), it comes with few theoretical guarantees; especially, a mathematical guarantee that the bi-level optimization problem (3) gives the minimum of the o.o.d. risk (1) has not yet been established.[2] The original IRM paper did not mention any theoretical properties for the minimum of (3). Rosenfeld et al. (2021) proved that, assuming that data follow a simple linear Gaussian structural equation model (SEM), a predictor obtained by (3) makes a prediction relying only on a feature of $X^e \in \mathcal{X}$ whose distribution does not depend on domains (Rosenfeld et al., 2021, Section 5). However, their analysis did not focus on relations between the bi-level optimization problem (3) and the o.o.d. risk (2). More recently, Kamath et al. (2021) provided an example of distributions on which a minimum of (3) does not minimize the o.o.d risk (Kamath et al., 2021, Section 4). However, their analysis assumed that data follow particular SEMs constructed to derive the case where (3) does not provide a minimum of the o.o.d. risk; for verifying the o.o.d. performance of the bi-leveled optimization problem (3), it should be analyzed under more general assumptions on distributions.

Aiming at providing a theoretical justification for IRM, this paper rigorously proves that a solution to the bi-leveled optimization problem (3) also minimizes the o.o.d. risk (1); formally speaking, we prove that the

---

[1]The definition is based on conditional independence (Peters et al., 2016; Koyama & Yamaguchi, 2021; Rojas-Carulla et al., 2018), while Arjovsky et al. (2019); Ahuja et al. (2020) used a different type of invariances based on $\arg\min_w \mathcal{R}^e(w \circ \Phi)$ instead of $P_{Y^e|\Phi(X^e)}$. Throughout the paper, we argue by adopting the definition based on conditional independence.

[2]Since it is difficult to solve the bi-leveled optimization problem (3), several papers have proposed optimization methods for (3) such as IRMv1 (Arjovsky et al., 2019) or Invariant Rationalization (Chang et al., 2020). While their optimization ability for solving (3) should also be discussed theoretically, this paper does not address it and only focuses on the problem of whether, assuming that (3) can be solved completely, the resulting predictor minimizes the o.o.d. risk.

inclusion

$$\arg\min_{\Phi\in\mathcal{I}_{tr},w\in\mathcal{W}} \sum_{e\in\mathcal{E}_{tr}} \mathcal{R}^e(w\circ\Phi) \subset \arg\min_{f\in\mathcal{F}} \mathcal{R}^{o.o.d.}(f) \tag{5}$$

is attained under certain conditions. The result also provides sufficient conditions on the training domains $\mathcal{E}_{tr}$ and the feature space $\mathcal{H}$ to minimize the o.o.d risk. In our analysis, we set distributions on domains $\mathcal{E}$ by the ones proposed in Rojas-Carulla et al. (2018). The distributions do not rely on any specific SEM structures, unlike existing theoretical analysis of IRM (Rosenfeld et al., 2021; Kamath et al., 2021), and they are used for the analysis of methods related to invariances (Rojas-Carulla et al., 2018; Toyota & Fukumizu, 2022).

The rest of the paper is organized as follows. Section 2 illustrates two main theorems. Section 2.1 provides the first main theorem, which states that the inclusion (5) is achieved in the regression case. In Section 2.2, we extend the first theorem to the classification case. The novelty and significance of these two theorems are discussed in Section 2.3. We provide a review of the prior works concerning the relationship between the bi-leveled optimization problem (3) and the o.o.d. risk (1) in Section 3. The two main theorems stated in Section 2 are proved in Section 4. Section 5 is devoted to brief concluding remarks.

## 2 Main Results

We explain the settings and assumptions persisting throughout our analysis.

We set domains $\{(X^e, Y^e)\}_{e\in\mathcal{E}}$ by the ones proposed in Rojas-Carulla et al. (2018). Let $\mathcal{X} := \mathcal{X}_1 \times \mathcal{X}_2$ where $\mathcal{X}_1 := \mathbb{R}^{d_1}$ and $\mathcal{X}_2 := \mathbb{R}^{d_2}$ with $d_1, d_2 \in \mathbb{N}_{>0}$, and $(X_1^I, Y^I)$ be a fixed random variable on $\mathcal{X}_1 \times \mathcal{Y}$. Rojas-Carulla et al. (2018) defined the domain set $\mathcal{E}$ by all the probability distributions with the fixed conditional distribution $P_{Y^I|X_1^I}$; namely, denoting $\Phi^{\mathcal{X}_1} : \mathcal{X} \to \mathcal{X}_1$ a projection onto $\mathcal{X}_1$, $\{(X^e, Y^e)\}_{e\in\mathcal{E}}$ is defined by

$$\{(X^e, Y^e)\}_{e\in\mathcal{E}} := \left\{ (X, Y) : \text{a random variable on } \mathcal{X} \times \mathcal{Y} \,\Big|\, P_{Y|\Phi^{\mathcal{X}_1}(X)} = P_{Y^I|X_1^I} \right\}. \tag{6}$$

Note that, under the setting (6), the projection $\Phi^{\mathcal{X}_1} : \mathcal{X} \to \mathcal{X}_1$ is an invariance among $\mathcal{E}$, because $P_{Y|\Phi^{\mathcal{X}_1}(X)} = P_{Y^I|X_1^I}$ for any $(X, Y) \in \{(X^e, Y^e)\}_{e\in\mathcal{E}}$. For simplicity of theoretical analysis, we assume that the conditional distribution $P_{Y^I|X_1^I}$ has a probability density function $p^I(y|x_1)$.

We explain assumptions about the feature space and models. The feature space $\mathcal{H}$ for an invariance $\Phi \in \mathcal{I}_{tr}$ is assumed to be the multi-dimensional Euclidean space $\mathbb{R}^{d_{\mathcal{H}}}$. Moreover, we assume that $\Phi \in \mathcal{I}_{tr}$ and $w \in \mathcal{W}$ in the minimization problem (3) run only continuous functions; namely, we investigate the property of a solution for

$$\min_{\Phi\in\mathcal{I}_{tr}^{\mathcal{C}_0}, w\in\mathcal{W}^{\mathcal{C}_0}} \sum_{e\in\mathcal{E}_{tr}} \mathcal{R}^e(w\circ\Phi), \tag{7}$$

where $\mathcal{W}^{\mathcal{C}_0}$ is the set of all *continuous* functions $w : \mathcal{H} \to \mathcal{Y}$, and $\mathcal{I}_{tr}^{\mathcal{C}_0}$ is the set of *continuous* invariances captured by a training domain $\mathcal{E}_{tr}$:

$$\mathcal{I}_{tr}^{\mathcal{C}_0} := \left\{ \Phi : \mathcal{X} \to \mathcal{H} \,|\, P_{Y^{e_1}|\Phi(X^{e_1})} = P_{Y^{e_2}|\Phi(X^{e_2})} \text{ for any } e_1, e_2 \in \mathcal{E}_{tr}, \ \Phi : \text{continuous} \right\}.$$

### 2.1 Case I: Least Square Loss

First, we consider the case where $\mathcal{Y} = \mathbb{R}^{d_{\mathcal{Y}}}$ $(d_{\mathcal{Y}} \in \mathbb{N}_{>0})$ and $l$ is the least square loss; that is, for a given predictor $f : \mathcal{X} \to \mathcal{Y}$, its risk $\mathcal{R}^e(f)$ on $(X^e, Y^e) \in \{(X^e, Y^e)\}_{e\in\mathcal{E}}$ is given by

$$\mathcal{R}^e(f) := \int \|y - f(x)\|^2 dP_{X^e, Y^e}.$$

The following theorem ensures that the optimization problem (7) provides a solution for the o.o.d. risk minimization problem (2) under four conditions:

**Theorem 1** (o.o.d. optimality of the bi-leveled optimization problem (7) under least square loss setting)**.** *Domains $\{(X^e, Y^e)\}_{e\in\mathcal{E}}$ are assumed to be (6). We also assume that the following four conditions hold:*

(i) $\mathcal{I}_{tr}^{\mathcal{C}_0} = \mathcal{I}^{\mathcal{C}_0}$, where $\mathcal{I}^{\mathcal{C}_0}$ is the set of continuous invariances captured by all domains $\mathcal{E}$, not training domains $\mathcal{E}_{tr}$:

$$\mathcal{I}^{\mathcal{C}_0} := \left\{ \Phi : \mathcal{X} \to \mathcal{H} \mid P_{Y^{e_1}|\Phi(X^{e_1})} = P_{Y^{e_2}|\Phi(X^{e_2})} \text{ for any } e_1, e_2 \in \mathcal{E}, \quad \Phi : \text{continuous} \right\}.$$

(ii) $\bigcup_{e \in \mathcal{E}_{tr}} supp(P_{\Phi^{\mathcal{X}_1}(X^e)}) = \mathcal{X}_1$. Here, for probability measure $\mu$ on $\mathcal{X}_1$, $supp(\mu)$ is defined by

$$supp(\mu) := \overline{\{x_1 \in \mathcal{X}_1 \mid N_{x_1} : open\ neighborhood\ around\ x_1 \Rightarrow \mu(N_{x_1}) > 0\}}.$$

(iii) The dimensions $d_1$ and $d_{\mathcal{H}}$ on the subspace $\mathcal{X}_1 \subset \mathcal{X}$ of the input space $\mathcal{X}$ and the feature space $\mathcal{H} = \mathbb{R}^{d_{\mathcal{H}}}$ satisfy $d_1 \le d_{\mathcal{H}}$.

(iv) $P_{Y^I|X_1^I}$ has a continuous probability density function $p^I(y|x_1)$. Here, we call $p^I(y|x_1)$ continuous when correspondence $\mathcal{X}_1 \times \mathcal{Y} \in (x_1, y) \longmapsto p^I(y|x_1)$ is continuous.

*Then, we have*

$$\arg\min_{\Phi \in \mathcal{I}_{tr}^{\mathcal{C}_0}, w \in \mathcal{W}^{\mathcal{C}_0}} \sum_{e \in \mathcal{E}_{tr}} \mathcal{R}^e(w \circ \Phi) \subset \arg\min_{f \in \mathcal{F}} \mathcal{R}^{o.o.d.}(f). \tag{8}$$

*Here, $\mathcal{F}$ is the set of all measurable functions $f : \mathcal{X} \to \mathcal{Y}$.*

We explain the feasibilities and interpretations of the above four conditions.

**Condition (i):** Condition (i) implies that invariances captured by training domains $\mathcal{E}_{tr}$ correspond to the ones by all domains $\mathcal{E}$. Arjovsky et al. (2019) also discussed the relationship between the equation $\mathcal{I}_{tr} = \mathcal{I}$ and o.o.d. generalization, briefly illustrating that the equation $\mathcal{I}_{tr} = \mathcal{I}$ facilitates the estimation of a predictor with high o.o.d. performance solely based on data from training domains $\mathcal{E}_{tr}$ (Arjovsky et al., 2019, Section 4.1). If it holds that $\mathcal{I}_{tr} = \mathcal{I}$, we can capture an invariance $\Phi \in \mathcal{I}$ among all domains $\mathcal{E}$ only using the training domains $\mathcal{E}_{tr}$. Arjovsky et al. (2019) pointed that, once an invariance $\Phi \in \mathcal{I}$ among all domains is obtained, a predictor $w^*$ that minimizes risks only on training domains $\mathcal{E}_{tr}$, namely $w^* \in \arg\min_{\mathcal{W}} \sum_{e \in \mathcal{E}_{tr}} \mathcal{R}^e(w \circ \Phi)$, satisfies $\mathcal{R}^e(w^* \circ \Phi) = \min_w \mathcal{R}^e(w \circ \Phi)$ for all domains $e \in \mathcal{E}$, including unseen domains $\mathcal{E} - \mathcal{E}_{tr}$, under certain settings. Developing the discussion by Arjovsky et al. (2019), Theorem 1 clarifies a more rigorous relation among the equation $\mathcal{I}_{tr} = \mathcal{I}$, the o.o.d. risk (1), and the bi-leveled optimization problem (7): the equation $\mathcal{I} = \mathcal{I}_{tr}$ is one of the sufficient conditions for the bi-leveled optimization problem (7) to minimize the o.o.d. risk (1).

The condition $\mathcal{I}_{tr} = \mathcal{I}$ is not generally satisfied and Peters et al. (2016); Arjovsky et al. (2019) presented sufficient conditions on the training domains $\mathcal{E}_{tr}$ for the equation $\mathcal{I}_{tr} = \mathcal{I}$ when data follow simple SEMs. Peters et al. (2016) proved the equation $\mathcal{I}_{tr} = \mathcal{I}$ holds when distributions on domains follow a linear Gaussian SEM and training data are obtained by certain types of interventions (Peters et al., 2016, Section 4.3). Arjovsky et al. (2019) generalized the result by Peters et al. (2016). Assuming that data follow a linear SEM, which is not restricted to a Gaussian distribution and a certain type of interventions, Arjovsky et al. (2019) deduced a sufficient condition for the equality $\mathcal{I}_{tr} = \mathcal{I}$ on training domains $\mathcal{E}_{tr}$, which is called *lying in the general position* (Arjovsky et al., 2019, Assumption 8). On the other hand, sufficient conditions for the equality $\mathcal{I}_{tr} = \mathcal{I}$ under the setting (6) have not yet been revealed. Providing them would be an important area for future research.

**Conditions (ii) and (iii):** As shown in Lemma 3, the conditional expectation $\int y \cdot p^I(y|x_1) dy = \mathbb{E}[Y^I = y|X_1^I = x_1]$ achieves the minimization of the o.o.d. risk, signifying that the information embedded in $\mathcal{X}_1$ is important for predicting response variables on unseen domains. Condition (ii) implies that the support of training domains $\mathcal{E}_{tr}$ covers $\mathcal{X}_1$ that contains such important information for o.o.d. prediction. Condition (iii) implies that $\mathcal{H}$ is such a large feature space that a feature $\Phi : \mathcal{X} \to \mathcal{H}$ can preserve information on the $\mathcal{X}_1$-component of $x \in \mathcal{X}$ by selecting $\Phi$ appropriately. Condition (iii) also provides a practical perspective on how to construct the feature space $\mathcal{H} = \mathbb{R}^{d_{\mathcal{H}}}$: the dimension $d_{\mathcal{H}}$ on the feature space $\mathcal{H}$ should be fixed high. The dimension $d_{\mathcal{H}}$ of the feature space is fixed by hand, and hence, Condition (iii) is expected to hold unless we fix the dimension of the feature space too small.

**Condition (iv):** Condition (iv) presents continuity of the p.d.f. of $P_{Y^I|X_1^I}$. By Condition (iv), we also have continuity of the conditional expectation $\int y \cdot p^I(y|x_1)dy = \mathbb{E}[Y^I = y|X_1^I = x_1]$. In our analysis, we assume that the model $\mathcal{W}^{\mathcal{C}_0}$ consists of all continuous functions, and hence, Condition (iv) ensures that the model includes the conditional expectation $\mathbb{E}[Y^I|X_1^I]$, which minimizes the o.o.d. risk (Lemma 3).

## 2.2 Case II: Cross Entropy Loss

Theorem 1 can be easily extended to the classification case where $w \in \mathcal{W}$ has a probabilistic output and evaluate risks by the cross entropy loss. Let $\mathcal{Y}$ be a finite set $\{1, ..., m\}$ ($m \in \mathbb{N}_{>0}$), and we model $w : \mathcal{H} \to \mathcal{Y}$ by $p_\theta : \mathcal{H} \to \mathcal{P}_{\mathcal{Y}}$, where $\mathcal{P}_{\mathcal{Y}}$ denotes the set of probabilities on $\mathcal{Y}$; namely

$$\mathcal{P}_{\mathcal{Y}} := \left\{ p \in \mathbb{R}_+^m \middle| \sum_{i=1}^m p_i = 1 \right\}.$$

Here, $\mathbb{R}_+ := \{x \in \mathbb{R} \,|\, x \geq 0\}$ and $p_i$ denotes the $i$-th component of $p$. We call $p_\theta : \mathcal{H} \to \mathcal{P}_{\mathcal{Y}}$ continuous, that is $p_\theta \in \mathcal{W}^{\mathcal{C}_0}$, when correspondence $\mathcal{H} \ni h \longmapsto p_\theta(h) \in \mathbb{R}^{|\mathcal{Y}|}$ is continuous, seeing $p_\theta(h) \in \mathcal{P}_{\mathcal{Y}}$ as a vector on $\mathbb{R}^{|\mathcal{Y}|}$. For a given $p_\theta : \mathcal{H} \to \mathcal{P}_{\mathcal{Y}}$ and $i \in \mathcal{Y}$, $(p_\theta(h))_i$ is often abbreviated by $p_\theta(i|h)$. The risk evaluated by the cross-entropy loss is then written as

$$\mathcal{R}^e(p_\theta \circ \Phi) = \int -\log p_\theta(Y^e|\Phi(X^e))dP_{X^e,Y^e}.$$

We expand Theorem 1 to the above classification case:

**Theorem 2** (o.o.d. optimality of the bi-leveled optimization problem (7) under cross-entropy loss setting)**.** *Domains* $\{(X^e, Y^e)\}_{e \in \mathcal{E}}$ *are assumed to be (6). Assume that, in addition to (i) $\sim$ (iii) in Theorem 1, the following condition (v) holds:*

*(v) For any $x_1^* \in \mathcal{X}_1$, $\#\left\{y \in \mathcal{Y} \,\middle|\, p^I(y|x_1^*) > 0\right\} > 1$.*

*Then, we have the inclusion*

$$\arg\min_{\Phi \in \mathcal{I}_{tr}^{\mathcal{C}_0}, p_\theta \in \mathcal{W}^{\mathcal{C}_0}} \sum_{e \in \mathcal{E}_{tr}} \mathcal{R}^e(p_\theta \circ \Phi) \subset \arg\min_{p_\theta \in \mathcal{F}} \mathcal{R}^{o.o.d.}(f), \tag{9}$$

*where $\mathcal{F}$ is the set of all measurable functions $f : \mathcal{X} \to \mathcal{P}_{\mathcal{Y}}$.[3]*

Condition (v) indicates that domains $\{(X^e, Y^e)\}_{e \in \mathcal{E}}$ have high uncertainty in labels $y \in \mathcal{Y}$ given $x_1 \in \mathcal{X}_1$. The condition is expected to be feasible when classes $\mathcal{Y}$ are subdivided and difficult to be uniquely determined from $x_1 \in \mathcal{X}_1$.

## 2.3 Novelty and Significance of Theorems 1 and 2

Theorems 1 and 2 and their proofs have the following four novel and significant points:

**Setting of Domains** The first point is the setting of domains. The setting by Rojas-Carulla et al. (2018), which is used throughout our analysis, does not impose any specific SEM structures, linearity, and Gaussianity on domains while existing works on theoretical analysis of IRM assumed that data follow simple SEMs. Theorems 1 and 2 indicate that, under such a general setting, IRM presents the minimum of the o.o.d. risk. This implies that our results provide a solid foundation to use IRM for a broad range of o.o.d. generalization problem.

---

[3]The same as the definition of *continuous*, we call $f \in \mathcal{F}$ measurable when correspondence $\mathcal{X} \ni x \longmapsto f(x) \in \mathbb{R}^{|\mathcal{Y}|}$ is measurable, seeing $f(x) \in P_{\mathcal{Y}}$ as a vector on $\mathbb{R}^{|\mathcal{Y}|}$.

**Assumption on the Underlying Distribution** $P_{Y^I|X_1^I}$   As well as the assumption on domains, prior theoretical results of IRM assume that the underlying true distribution $P_{Y^I|X_1^I}$ is represented by a simple SEM (Arjovsky et al., 2019; Rosenfeld et al., 2021; Kamath et al., 2021). On the other hand, Condition (iv) only imposes $P_{Y^I|X_1^I}$ on the continuity; hence, Condition (iv) is a significantly mild condition in comparison with the assumptions on $P_{Y^I|X_1^I}$ by prior works.

**Characterization of Invariance**   Second, a theoretical characterization of invariances $\Phi^* \in \mathcal{I}^{\mathcal{C}_0}$ is given in Lemmas 4 and 6: it is proved that $\Phi^* \in \mathcal{I}^{\mathcal{C}_0}$ can be represented as $\Phi^* = \Psi^* \circ \Phi^{\mathcal{X}_1}$ for some continuous map $\Psi^*$. Any theoretical characterizations have not yet been presented, and hence, the results in Lemmas 4 and 6 are novel. To present the non-trivial characterization, we develop a novel theoretical technique based on the proof by contradiction. Lemmas 4 and 6 play an important role in our desirable assertion (5), and hence, the derivation of these lemmas is a significant technical contribution of our analysis.

**Range of Invariance**   The fourth point is a range of invariances $\Phi$: we assume that $\Phi$ run all continuous functions, while most of the existing works on theoretical analysis of IRM assume that $\Phi$ run more simplified functions, such as linear functions (Rosenfeld et al., 2021) or variable selections (Toyota & Fukumizu, 2022). It is common to construct a learning model of invariances with deep neural networks in the context of IRM, and hence, the variable selection and linear function settings by Toyota & Fukumizu (2022); Rosenfeld et al. (2021) are significantly simplified to analyze IRM. On the other hand, our large class of continuous functions is relatively realistic compared to existing ones, since it is widely recognized that neural networks of sufficient size can represent a wide range of functions (Cybenko, 1989; Hornik et al., 1989; Barron, 1993; Mhaskar, 1996; Sonoda & Murata, 2017).

## 3   Previous Works

As explained in Section 1, Rosenfeld et al. (2021); Kamath et al. (2021) derived the theoretical results concerning the minimum of the bi-leveled optimization problem (3) and its connection to the o.o.d. risk (1). Rosenfeld et al. (2021) proved that a predictor obtained by minimizing (3) predicts $Y^e \in \mathcal{Y}$ relying only on a feature of $X^e \in \mathcal{X}$ whose distribution does not depend on domains (Rosenfeld et al., 2021, Section 5). However, they did not provide any connections between the minimum of (3) and the o.o.d risk. Moreover, they assume that data follow a linear Gaussian SEM, and that invariances $\Phi$ in the bi-leveled optimization problem run linear functions for simplicity. Unlike their analysis, this paper derives the direct relations between (3) and the o.o.d. risk (1). Additionally, we assume that data follow the distributions proposed by Rojas-Carulla et al. (2018) that do not rely on any specific SEM structures and that invariances run all continuous functions including neural networks. Kamath et al. (2021) provided an example of distributions on which a minimum of (3) does not minimize the o.o.d risk. However, the distributions are particular SEMs constructed to derive the case where (5) is violated, and analysis in more general settings is required (Kamath et al., 2021, Section 4). In construct, the distributions by Rojas-Carulla et al. (2018) used in this paper do not rely on any specific SEM structures, and they are used to analyze estimation methods related to invariances (Rojas-Carulla et al., 2018; Toyota & Fukumizu, 2022).

Arjovsky et al. (2019); Koyama & Yamaguchi (2021); Rojas-Carulla et al. (2018) discussed theoretical relations between invariances and the o.o.d. risk (1). As explained in the last section, Arjovsky et al. (2019) intuitively explained that the condition $\mathcal{I}_{tr} = \mathcal{I}$ facilitates an estimation of a predictor which can predict $Y^e$ on unseen domains only by data from training domains $\mathcal{E}_{tr}$ (Arjovsky et al., 2019, Section 4.1). They also derived sufficient conditions on training domains for the equation $\mathcal{I}_{tr} = \mathcal{I}$, assuming that data follow a simple linear SEM (Arjovsky et al., 2019, Theorem 9). Koyama & Yamaguchi (2021); Rojas-Carulla et al. (2018) presented sufficient conditions for an invariance $\Phi$ to achieve the minimum of (1). Koyama & Yamaguchi (2021) proved that the invariance that maximizes the mutual information with labels also maximizes the o.o.d. risk. Rojas-Carulla et al. (2018) proved that, under the domain setting (6), the conditional expectation $\mathbb{E}[Y^e|\Phi^{\mathcal{X}_1}(X^e) = x_1]$ also minimizes the o.o.d. risk, even when $\mathbb{E}[Y^e|\Phi^{\mathcal{X}_1}(X^e)]$ is nonlinear. However, all the results by Koyama & Yamaguchi (2021); Rojas-Carulla et al. (2018); Arjovsky et al. (2019) did not deal with any theoretical connections between invariances obtained by minimizing the bi-leveled optimization problem (3) and the o.o.d. risk (1). It does not follow obviously that the minimum of (3) satisfies these sufficient

conditions by Koyama & Yamaguchi (2021); Rojas-Carulla et al. (2018), and hence our main theorems can not be deduced as a trivial corollary of the results by Koyama & Yamaguchi (2021); Rojas-Carulla et al. (2018). To discuss the non-trivial relation between the bi-leveled optimization problem (3) and the o.o.d. risk (1), we establish a novel characterization of invariances (Lemmas 4 and 6), and derive the main theorems based on it.

To reduce the annotation cost required for the original IRM approach, Toyota & Fukumizu (2022) introduced a new bi-level optimization problem similar to (3). They considered a situation in which the training data for target classification are provided in only one domain, while the task of a higher label hierarchy, which requires lower annotation cost, has data from multiple domains. Under the availability of data, they deduced a bi-level optimization problem, in which invariances were given by additional data in a higher label hierarchy. For further details, we refer the reader to the original paper Toyota & Fukumizu (2022). Their study provided a detailed theoretical analysis concerning their method and its connection to the o.o.d. risk; however, they did not analyze relationships between their bi-leveled optimization problem and the o.o.d. risk, which is the focus of this paper. Instead, they investigated relationships between an optimization method for their bi-level optimization problem and the o.o.d. risk. Moreover, they assume that invariances $\Phi$ run all variable selections for simplicity of theoretical analysis. On the other hand, this paper derives the direct relations between the minimum of the bi-leveled optimization problem and the o.o.d. risk (1). Moreover, we consider the more realistic setting for the analysis of IRM where invariances $\Phi$ run all continuous functions.

## 4  Proofs

In this section, we prove Theorem 1 and 2. Through the section, for $X^e \in \mathcal{X}$ and $x \in \mathcal{X}$, its $\mathcal{X}_i$-components $(i = 1, 2)$ are denoted by $X_i^e$ and $x_i$ respectively.

### 4.1  Proof Sketch of Main Theorems

Before giving rigorous proof, we briefly describe the rough proof sketch of the main theorems. The following two lemmas (A) and (B) play an important role in our proof:

(A) The conditional expectation $\mathbb{E}[Y^e|X_1^e] = \mathbb{E}[Y^I|X_1^I]$ and conditional probability $P_{Y^e|X_1^e} = P_{Y^I|X_1^I}$ minimize the o.o.d. risk under the least-square and cross-entropy losses respectively (Lemmas 3 and 5).

(B) Any invariance among all domains can be represented by the composition of the projection onto $\mathcal{X}_1$; that is, $\Phi \in \mathcal{I}^{\mathcal{C}_0}$ can be represented as

$$\Phi = \Psi \circ \Phi^{\mathcal{X}_1}$$

for some continuous map $\Psi$ (Lemmas 4 and 6).

The two lemmas intuitively conclude the proof of the main theorem as follows. Firstly, since $\mathcal{I}_{tr}^{\mathcal{C}_0} = \mathcal{I}^{\mathcal{C}_0}$ holds (Condition (i)), observe that a predictor in (3) runs composition maps $w \circ \Phi$ with $w \in \mathcal{W}^{\mathcal{C}_0}$ and $\Phi \in \mathcal{I}^{\mathcal{C}_0}$. Moreover, since the above second lemma (B) ensures that $\Phi \in \mathcal{I}^{\mathcal{C}_0}$ can be represented by the composition of the projection onto $\mathcal{X}_1$, we can see that a predictor in (3) runs $w \circ \Phi^{\mathcal{X}_1}$ for some function class $w \in \mathcal{W}^*$, and hence, the bi-leveled optimization problem is expressed as

$$\min_{w \in \mathcal{W}^*} \sum_{e \in \mathcal{E}_{tr}} \mathcal{R}^e(w \circ \Phi^{\mathcal{X}_1}). \tag{10}$$

It is well-known that, assuming that $w$ runs all measurable functions,

$$\hat{w} \in \underset{w}{\arg\min}\, \mathcal{R}^e(w \circ \Phi^{\mathcal{X}_1}) \iff \hat{w}(x_1) = \mathbb{E}[Y^e|X_1^e = x_1] = \mathbb{E}[Y^I|X_1^I = x_1] \quad P_{X_1^e} - \text{almost everywhere}.$$

or

$$\hat{w} \in \underset{w}{\arg\min}\, \mathcal{R}^e(w \circ \Phi^{\mathcal{X}_1}) \iff \hat{w}(x_1) = P_{Y^e|X_1^e=x_1} = P_{Y^I|X_1^I=x_1} \quad P_{X_1^e} - \text{almost everywhere}.$$

hold for any $e \in \mathcal{E}$ under the least-square and cross-entropy losses respectively (Christmann & Steinwart, 2008, Example 2.6). Hence, ignoring the capability of $\mathcal{W}^*$ and the discussion of *almost everywhere*, we can see that

$$\mathbb{E}[Y^I|X_1^I = x_1] \approx \arg\min_{w \in \mathcal{W}^*} \sum_{e \in \mathcal{E}_{tr}} \mathcal{R}^e(w \circ \Phi^{\mathcal{X}_1}) \tag{11}$$

or

$$P_{Y^I|X_1^I = x_1} \approx \arg\min_{w \in \mathcal{W}^*} \sum_{e \in \mathcal{E}_{tr}} \mathcal{R}^e(w \circ \Phi^{\mathcal{X}_1}) \tag{12}$$

hold. Combining eq.s (11), (12) and the first lemma (A), it concludes the main theorems intuitively. In the following section, we give the rigorous justification of the above rough proof sketch.

### 4.2 Proof of Theorem 1

To prove the main theorem, we prepare two lemmas.

**Lemma 3.** *Let $w^I : \mathcal{X}_1 \to \mathcal{Y}$ be the conditional expectation obtained by $p^I(y|x_1)$; namely,*

$$w^I(x_1) = \mathbb{E}[Y^I|X_1^I = x_1] := \int y \cdot p^I(y|x_1) dy.$$

*Then,*

$$w^I \circ \Phi^{\mathcal{X}_1} \in \arg\min_{f:\mathcal{X} \to \mathcal{Y}} \mathcal{R}^{o.o.d.}(f).$$

**Lemma 4.** *Any $\Phi \in \mathcal{I}_{tr}^{C^0}$ is represented as*

$$\Phi = \Psi \circ \Phi^{\mathcal{X}_1}$$

*for some continuous map $\Psi : \mathcal{X}_1 \to \mathcal{H}$.*

**Proof of Lemma 3** [4] It suffices to prove the following statement:

For any $f \in \mathcal{F}$ and $(X^a, Y^a) \in \{(X^e, Y^e)\}_{e \in \mathcal{E}}$, there exists $(X^b, Y^b) \in \{(X^e, Y^e)\}_{e \in \mathcal{E}}$ such that

$$\int \|w^I \circ \Phi^{\mathcal{X}_1}(x) - y\|^2 dP_{X^a, Y^a}(x, y) \le \int \|f(x) - y\|^2 dP_{X^b, Y^b}(x, y). \tag{13}$$

Take arbitrary $f \in \mathcal{F}$ and $(X^a, Y^a) \in \{(X^e, Y^e)\}_{e \in \mathcal{E}}$. Define $(X^b, Y^b) \in \{(X^e, Y^e)\}_{e \in \mathcal{E}}$ such that its distribution is the direct product $P_{X_1^a, Y^a} \otimes P_{X_2}$, where $P_{X_1^a, Y^a}$ is the marginal distribution of $P_{X^a, Y^a}$ on $\mathcal{X}_1 \times \mathcal{Y}$ and $P_{X_2}$ is an arbitrary distribution on $\mathcal{X}_2$.

Then, the right-hand side of the inequality (13) is given by

$$\int \|f(x) - y\|^2 dP_{X^b, Y^b}(x, y) = \int \|f(x) - y\|^2 d(P_{X_1^a, Y^a} \otimes P_{X_2})(x, y)$$

$$= \int P_{X_2}(x_2) \int \|f(x_1, x_2) - y\|^2 dP_{X_1^a, Y^a}(x_1, y).$$

Clearly, for any $x_2^* \in \mathcal{X}_2$, the inequality

$$\int \|f(x_1, x_2^*) - y\|^2 dP_{X_1^a, Y^a}(x_1, y) \ge \int \|\mathbb{E}[Y^{e_1}|X_1^a = x_1] - y\|^2 dP_{X_1^a, Y^a}(x_1, y)$$

---

[4]The proof is essentially the same as the one for Theorem 1 in Rojas-Carulla et al. (2018) and Theorem 6 in Toyota & Fukumizu (2022).

holds, because the minimum of a risk on the least square loss is attained at the conditional expectation $\mathbb{E}[Y^a|X_1^a]$. Hence, we obtain

$$
\begin{aligned}
\int \|f(x) - y\|^2 dP_{X^b, Y^b}(x, y) &= \int P_{X_2}(x_2) \int \|f(x_1, x_2) - y\|^2 dP_{X_1^a, Y^a}(x_1, y) \\
&\geq \int P_{X_2}(x_2) \int \|\mathbb{E}[Y^{e_1}|X_1^a = x_1] - y\|^2 dP_{X_1^a, Y^a}(x_1, y) \\
&= \int \|\mathbb{E}[Y^a|X_1^a = x_1] - y\|^2 dP_{X_1^a, Y^a}(x_1, y) \\
&= \int P_{X_2^a|X_1^a, Y^a}(x_2) \int \|\mathbb{E}[Y^a|X_1^a = x_1] - y\|^2 dP_{X_1^a, Y^a}(x_1, y) \\
&= \int \|\mathbb{E}[Y^a|X_1^a = \Phi^{\mathcal{X}_1}(x)] - y\|^2 dP_{X^a, Y^a}(x, y) \\
&= \int \|w^I \circ \Phi^{\mathcal{X}_1}(x) - y\|^2 dP_{X^a, Y^a}(x, y),
\end{aligned}
$$

which concludes the proof. Here, the last equality is derived from the fact that the conditional expectation $\mathbb{E}[Y^e|X_1^e = \Phi^{\mathcal{X}_1}(x)]$ does not depend on $e \in \mathcal{E}$ and corresponds to $w^I \circ \Phi^{\mathcal{X}_1}$. $\qquad\square$

**Proof sketch of Lemma 4**  Before providing a complete proof, we show a proof sketch of Lemma 4 to make the flow of our proof easier to understand. First, we prove that $\Phi \in \mathcal{I}_{tr}^{C^0}$ can be represented as

$$
\Phi = \Psi \circ \Phi^{\mathcal{X}_1} \tag{14}
$$

by some map $\Psi : \mathcal{X}_1 \to \mathcal{H}$, which is not restricted to a continuous map. Take arbitrary $\Phi \in \mathcal{I}_{tr}^{C^0}$. Then, since $\Phi \in \mathcal{I}^{C^0} = \mathcal{I}_{tr}^{C^0}$ (Condition (i)), for any $(X^a, Y^a), (X^b, Y^b) \in \{(X^e, Y^e)\}_{e \in \mathcal{E}}$,

$$
P_{Y^a|\Phi(X^a)} = P_{Y^b|\Phi(X^b)},
$$

and therefore, we have

$$
P_{Y^a|\Phi(X^a)}(N|\Phi(x)) = P_{Y^b|\Phi(X^b)}(N|\Phi(x)) \tag{15}
$$

for any set $N \subset \mathcal{Y}$ and $\forall x \in \mathcal{X}$. We prove the statement (14) by contradiction. Assume that there exist no maps $\Psi$ that satisfy (14). Then, there exist $x_1^* \in \mathcal{X}_1$, $x_2^*, x_2^{**} \in \mathcal{X}_2$ such that

$$
\Phi(x_1^*, x_2^*) \neq \Phi(x_1^*, x_2^{**}).^5
$$

By utilizing $x_1^* \in \mathcal{X}_1$, $x_2^*, x_2^{**} \in \mathcal{X}_2$, we can construct $(X^a, Y^a), (X^b, Y^b) \in \{(X^e, Y^e)\}_{e \in \mathcal{E}}$ and $N \subset \mathcal{Y}$ which satisfy

$$
P_{Y^a|\Phi(X^a)}(N|\Phi(x_1^*, x_2^*)) \neq P_{Y^b|\Phi(X^b)}(N|\Phi(x_1^*, x_2^*)).
$$

This contradicts the assumption (15), and we can conclude $\Phi \in \mathcal{I}_{tr}^{C^0}$ can be represented as (14). The continuity of $\Psi$ is easily derived from the continuity of $\Phi$, and we can conclude the proof.

**Proof of Lemma 4**  First, we prove that $\Phi \in \mathcal{I}_{tr}^{C^0}$ can be represented as

$$
\Phi = \Psi \circ \Phi^{\mathcal{X}_1} \tag{16}
$$

by some map $\Psi : \mathcal{X}_1 \to \mathcal{H}$, which is not restricted to a continuous map. We prove this statement by contradiction. Take $\Phi \in \mathcal{I}_{tr}^{C^0}$ and assume that there exist no maps $\Psi$ which satisfy (16). Then, there exist $x_1^* \in \mathcal{X}_1$, $x_2^*, x_2^{**} \in \mathcal{X}_2$ such that

$$
\Phi(x_1^*, x_2^*) \neq \Phi(x_1^*, x_2^{**}). \tag{17}
$$

---

[5] If $\Phi(x_1^*, x_2^*) = \Phi(x_1^*, x_2^{**})$ for any $x_1^* \in \mathcal{X}_1$, $x_2^*, x_2^{**} \in \mathcal{X}_2$, $\Phi$ depend only on the first component $\mathcal{X}_1$; hence, we can see that $\Phi \in \mathcal{I}_{tr}^{C^0}$ can be represented as $\Phi = \Psi \circ \Phi^{\mathcal{X}_1}$ by some map $\Psi$, which contradicts to the assumption.

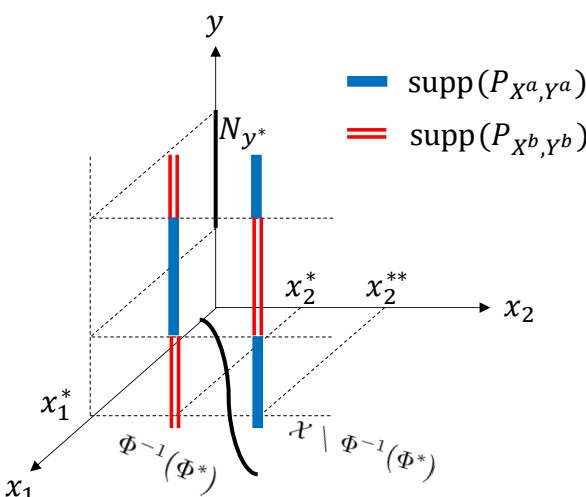

Figure 1: Supports of probability distributions $P_{X^a,Y^a}$ and $P_{X^b,Y^b}$. The figure implies that $P_{X^a,Y^a}(N_{y^*} \times \Phi^{-1}(\Phi^*)) \neq 0$ and $P_{X^b,Y^b}(N_{y^*} \times \Phi^{-1}(\Phi^*)) = 0$ ($\because (x_1^*, x_2^{**}) \notin \Phi^{-1}(\Phi^*)$ (17)), and that $P_{X^a,Y^a}(\Phi^{-1}(\Phi^*)) \neq 0$ and $P_{X^b,Y^b}(\Phi^{-1}(\Phi^*)) \neq 0$. These Eqs. lead us $P_{Y^a|\Phi(X^a)}(N_{y^*}|\Phi^*) \neq 0 = P_{Y^b|\Phi(X^b)}(N_{y^*}|\Phi^*)$.

Fix $y^* \in \mathcal{Y}$ with $p^I(y^*|x_1^*) > 0$ and take an open neighborhood $N_{y^*} \subset \mathcal{Y}$ centered at $y^*$ which satisfies

$$0 < \int_{N_{y^*}} p^I(y|x_1^*) dy < 1.$$

Here, the existence of $N_{y^*}$ is derived from the continuity of $p^I(\cdot|x_1^*)$ (Condition (iv)).

Define two maps $g^i : \mathcal{Y} \to \mathcal{X}_2$ ($i = 1, 2$) by

$$g^1(y) = \begin{cases} x_2^* & (y \in N_{y^*}) \\ x_2^{**} & (\text{ else }) \end{cases} \qquad g^2(y) = \begin{cases} x_2^{**} & (y \in N_{y^*}) \\ x_2^* & (\text{ else }). \end{cases}$$

Take two distributions $(X^a, Y^a), (X^b, Y^b) \in \{(X^e, Y^e)\}_{e \in \mathcal{E}}$ such that their distributions $P_{X^a,Y^a}$ and $P_{X^b,Y^b}$ coincide with

$$P_{X^a,Y^a} = P_{X_2^a|Y^a} \otimes P_{Y^I|X_1^I} \otimes P_{X_1}, \qquad P_{X^b,Y^b} = P_{X_2^b|Y^b} \otimes P_{Y^I|X_1^I} \otimes P_{X_1}.$$

Here,

- $P_{X_1}$ is a distribution on $\mathcal{X}_1$ where its p.d.f. coincides with a delta function $\delta_{x_1^*}(x_1)$ on $x_1^*$,

- the conditional p.d.f.s of $P_{X_2^a|Y^a}(\cdot|y)$ and $P_{X_2^b|Y^b}(\cdot|y)$ coincide with $\delta_{g^1(y)}(x_2)$ and $\delta_{g^2(y)}(x_2)$ respectively.

The supports of $P_{X^a,Y^a}$ and $P_{X^b,Y^b}$ are visualized in Fig. 1. As $\Phi \in \mathcal{I}^{\mathcal{C}_0} = \mathcal{I}_{tr}^{\mathcal{C}_0}$ (Condition (i)) and $(X^a, Y^a), (X^b, Y^b) \in \{(X^e, Y^e)\}_{e \in \mathcal{E}}$,

$$P_{Y^a|\Phi(X^a)}(N_{y^*}|\Phi^*) = P_{Y^b|\Phi(X^b)}(N_{y^*}|\Phi^*), \tag{18}$$

where $\Phi^* := \Phi(x_1^*, x_2^*)$. Let us compute $P_{Y^a|\Phi(X^a)}(N_{y^*}|\Phi^*)$ and $P_{Y^b|\Phi(X^b)}(N_{y^*}|\Phi^*)$ to derive $P_{Y^a|\Phi(X^a)}(N_{y^*}|\Phi^*) \neq P_{Y^b|\Phi(X^b)}(N_{y^*}|\Phi^*)$, which contradicts to the equality (18)[6]. We evaluate

$$P_{Y^a|\Phi(X^a)}(N_{y^*}|\Phi^*) = \frac{P_{\Phi(X^a),Y^a}(\{\Phi^*\} \times N_{y^*})}{P_{\Phi(X^a)}(\{\Phi^*\})} = \frac{P_{X^a,Y^a}(\Phi^{-1}(\Phi^*) \times N_{y^*})}{P_{X^a}(\Phi^{-1}(\Phi^*))}$$

---

[6]Fig. 1 illustrates the intuitive reason why $P_{Y^a|\Phi(X^a)}(N_{y^*}|\Phi^*) \neq P_{Y^b|\Phi(X^b)}(N_{y^*}|\Phi^*)$ is derived, and hence, will help us understand the following rigorous proof.

by computing its numerator and denominator separately. First, the numerator is evaluated as

$$P_{X^a,Y^a}(\Phi^{-1}(\Phi^*) \times N_{y^*}) = \int_{\Phi^{-1}(\Phi^*) \times N_{y^*}} \delta_{g^1(y)}(x_2) \cdot p^I(y|x_1) \cdot \delta_{x_1^*}(x_1) dx dy$$

$$= \int_{N_{y^*}} dy \int_{\Phi^{-1}(\Phi^*)} \delta_{g^1(y)}(x_2) \cdot p^I(y|x_1) \cdot \delta_{x_1^*}(x_1) dx.$$

Noting that $g^1(y) = x_2^*$ for $\forall y \in N_{y^*}$ and $\delta_{x_1^*}(x_1) \times \delta_{x_2^*}(x_2) = \delta_{(x_1^*,x_2^*)}(x_1, x_2)$, we obtain

$$\int_{N_{y^*}} dy \int_{\Phi^{-1}(\Phi^*)} \delta_{g^1(y)}(x_2) \cdot p^I(y|x_1) \cdot \delta_{x_1^*}(x_1) dx = \int_{N_{y^*}} dy \int_{\Phi^{-1}(\Phi^*)} \delta_{x_2^*}(x_2) \cdot p^I(y|x_1) \cdot \delta_{x_1^*}(x_1) dx$$

$$= \int_{N_{y^*}} dy \int_{\Phi^{-1}(\Phi^*)} \delta_{(x_1^*,x_2^*)}(x_1, x_2) \cdot p^I(y|x_1) dx.$$

Since $(x_1^*, x_2^*) \in \Phi^{-1}(\Phi^*)$, we have

$$\int_{N_{y^*}} dy \int_{\Phi^{-1}(\Phi^*)} \delta_{(x_1^*,x_2^*)}(x_1, x_2) \cdot p^I(y|x_1) dx = \int_{N_{y^*}} p^I(y|x_1^*) dy,$$

which leads us to the equality

$$P_{X^a,Y^a}(\Phi^{-1}(\Phi^*) \times N_{y^*}) = \int_{N_{y^*}} p^I(y|x_1^*) dy.$$

Next, let us evaluate the denominator $P_{X^a}(\Phi^{-1}(\Phi^*))$.

$$P_{X^a}(\Phi^{-1}(\Phi^*)) = P_{X^a,Y^a}(\Phi^{-1}(\Phi^*) \times \mathcal{Y}) = \int_{\Phi^{-1}(\Phi^*) \times \mathcal{Y}} \delta_{g^1(y)}(x_2) \cdot p^I(y|x_1) \cdot \delta_{x_1^*}(x_1) dx dy$$

$$= \int_{\mathcal{Y}} dy \int_{\Phi^{-1}(\Phi^*)} \delta_{g^1(y)}(x_2) \cdot p^I(y|x_1) \cdot \delta_{x_1^*}(x_1) dx$$

$$= \int_{N_{y^*}} dy \int_{\Phi^{-1}(\Phi^*)} \delta_{g^1(y)}(x_2) \cdot p^I(y|x_1) \cdot \delta_{x_1^*}(x_1) dx$$

$$+ \int_{\mathcal{Y}-N_{y^*}} dy \int_{\Phi^{-1}(\Phi^*)} \delta_{g^1(y)} \cdot p^I(y|x_1) \cdot \delta_{x_1^*}(x_1) dx$$

$$= \int_{N_{y^*}} dy \int_{\Phi^{-1}(\Phi^*)} \delta_{x_2^*}(x_2) \cdot p^I(y|x_1) \cdot \delta_{x_1^*}(x_1) dx$$

$$+ \int_{\mathcal{Y}-N_{y^*}} dy \int_{\Phi^{-1}(\Phi^*)} \delta_{x_2^{**}}(x_2) \cdot p^I(y|x_1) \cdot \delta_{x_1^*}(x_1) dx$$

$$= \int_{N_{y^*}} dy \int_{\Phi^{-1}(\Phi^*)} \delta_{(x_1^*,x_2^*)}(x_1, x_2) \cdot p^I(y|x_1) dx$$

$$+ \int_{\mathcal{Y}-N_{y^*}} dy \int_{\Phi^{-1}(\Phi^*)} \delta_{(x_1^*,x_2^{**})}(x_1, x_2) \cdot p^I(y|x_1) dx.$$

Here, the fourth equality is derived from the facts that $g^1(y) = x_2^*$ for $\forall y \in N_{y^*}$ and $g^1(y) = x_2^{**}$ for $\forall y \in \mathcal{Y} - N_{y^*}$. Noting that $(x_1^*, x_2^*) \in \Phi^{-1}(\Phi^*)$ and $(x_1^*, x_2^{**}) \notin \Phi^{-1}(\Phi^*)$, we obtain

$$P_{X^a}(\Phi^{-1}(\Phi^*)) = \int_{N_{y^*}} dy \int_{\Phi^{-1}(\Phi^*)} \delta_{(x_1^*,x_2^*)}(x_1, x_2) \cdot p^I(y|x_1) dx$$

$$+ \int_{\mathcal{Y}-N_{y^*}} dy \int_{\Phi^{-1}(\Phi^*)} \delta_{(x_1^*,x_2^{**})}(x_1, x_2) \cdot p^I(y|x_1) dx$$

$$= \int_{N_{y^*}} dy \cdot p^I(y|x_1^*) + \int_{\mathcal{Y}-N_{y^*}} dy \cdot 0$$

$$= \int_{N_{y^*}} p^I(y|x_1^*) dy.$$

Hence, we obtain

$$P_{Y^a|\Phi(X^a)}(N_{y^*}|\Phi^*) = \frac{P_{X^a,Y^a}(\Phi^{-1}(\Phi^*) \times N_{y^*})}{P_{X^a}(\Phi^{-1}(\Phi^*))}$$

$$= \frac{\int_{N_{y^*}} p^I(y|x_1^*)dy}{\int_{N_{y^*}} p^I(y|x_1^*)dy} = 1.$$

Next, let us evaluate

$$P_{Y^b|\Phi(X^b)}(N_{y^*}|\Phi^*) = \frac{P_{\Phi(X^b),Y^b}(\{\Phi^*\} \times N_{y^*})}{P_{\Phi(X^b)}(\{\Phi^*\})} = \frac{P_{X^b,Y^b}(\Phi^{-1}(\Phi^*) \times N_{y^*})}{P_{X^b}(\Phi^{-1}(\Phi^*))}.$$

The numerator is evaluated as

$$P_{X^b,Y^b}(\Phi^{-1}(\Phi^*) \times N_{y^*}) = \int_{\Phi^{-1}(\Phi^*) \times N_{y^*}} \delta_{g^2(y)}(x_2) \cdot p^I(y|x_1) \cdot \delta_{x_1^*}(x_1)dxdy$$

$$= \int_{N_{y^*}} dy \int_{\Phi^{-1}(\Phi^*)} \delta_{g^2(y)}(x_2) \cdot p^I(y|x_1) \cdot \delta_{x_1^*}(x_1)dx.$$

Noting that $g^2(y) = x_2^{**}$ for $\forall y \in N_{y^*}$, we obtain

$$\int_{N_{y^*}} dy \int_{\Phi^{-1}(\Phi^*)} \delta_{g^2(y)}(x_2) \cdot p^I(y|x_1) \cdot \delta_{x_1^*}(x_1)dx = \int_{N_{y^*}} dy \int_{\Phi^{-1}(\Phi^*)} \delta_{x_2^{**}}(x_2) \cdot p^I(y|x_1) \cdot \delta_{x_1^*}(x_1)dx$$

$$= \int_{N_{y^*}} dy \int_{\Phi^{-1}(\Phi^*)} \delta_{(x_1^*,x_2^{**})}(x_1,x_2) \cdot p^I(y|x_1)dx$$

$$= \int_{N_{y^*}} dy \cdot 0 = 0.$$

Here, the third equality is derived from $(x_1^*, x_2^{**}) \notin \Phi^{-1}(\Phi^*)$. Next, the denominator $P_{X^b}(\Phi^{-1}(\Phi^*))$ is evaluated as

$$P_{X^b}(\Phi^{-1}(\Phi^*)) = P_{X^b,Y^b}(\Phi^{-1}(\Phi^*) \times \mathcal{Y}) = \int_{\Phi^{-1}(\Phi^*) \times \mathcal{Y}} \delta_{g^2(y)}(x_2) \cdot p^I(y|x_1) \cdot \delta_{x_1^*}(x_1)dxdy$$

$$= \int_{\mathcal{Y}} dy \int_{\Phi^{-1}(\Phi^*)} \delta_{g^2(y)}(x_2) \cdot p^I(y|x_1) \cdot \delta_{x_1^*}(x_1)dx$$

$$= \int_{N_{y^*}} dy \int_{\Phi^{-1}(\Phi^*)} \delta_{g^2(y)}(x_2) \cdot p^I(y|x_1) \cdot \delta_{x_1^*}(x_1)dx +$$

$$\int_{\mathcal{Y}-N_{y^*}} dy \int_{\Phi^{-1}(\Phi^*)} \delta_{g^2(y)} \cdot p^I(y|x_1) \cdot \delta_{x_1^*}(x_1)dx$$

$$= \int_{N_{y^*}} dy \int_{\Phi^{-1}(\Phi^*)} \delta_{x_2^{**}}(x_2) \cdot p^I(y|x_1) \cdot \delta_{x_1^*}(x_1)dx$$

$$+ \int_{\mathcal{Y}-N_{y^*}} dy \int_{\Phi^{-1}(\Phi^*)} \delta_{x_2^*}(x_2) \cdot p^I(y|x_1) \cdot \delta_{x_1^*}(x_1)dx$$

$$= \int_{N_{y^*}} dy \int_{\Phi^{-1}(\Phi^*)} \delta_{(x_1^*,x_2^{**})}(x_1,x_2) \cdot p^I(y|x_1)dx$$

$$+ \int_{\mathcal{Y}-N_{y^*}} dy \int_{\Phi^{-1}(\Phi^*)} \delta_{(x_1^*,x_2^*)}(x_1,x_2) \cdot p^I(y|x_1)dx.$$

Noting that $(x_1^*, x_2^*) \in \Phi^{-1}(\Phi^*)$ and $(x_1^*, x_2^{**}) \notin \Phi^{-1}(\Phi^*)$, we obtain

$$
\begin{aligned}
P_{X^b}(\Phi^{-1}(\Phi^*)) &= \int_{N_{y^*}} dy \int_{\Phi^{-1}(\Phi^*)} \delta_{(x_1^*, x_2^{**})}(x_1, x_2) \cdot p^I(y|x_1) dx \\
&\quad + \int_{\mathcal{Y} - N_{y^*}} dy \int_{\Phi^{-1}(\Phi^*)} \delta_{(x_1^*, x_2^*)}(x_1, x_2) \cdot p^I(y|x_1) dx \\
&= \int_{N_{y^*}} dy \cdot 0 + \int_{\mathcal{Y} - N_{y^*}} dy \cdot p^I(y|x_1^*) \\
&= \int_{\mathcal{Y} - N_{y^*}} p^I(y|x_1^*) dy \neq 0.
\end{aligned}
$$

Hence, we obtain

$$
\begin{aligned}
P_{Y^b|\Phi(X^b)}(N_{y^*}|\Phi^*) &= \frac{P_{X^b, Y^b}(\Phi^{-1}(\Phi^*) \times N_{y^*})}{P_{X^b}(\Phi^{-1}(\Phi^*))} \\
&= \frac{0}{\int_{\mathcal{Y} - N_{y^*}} p^I(y|x_1^*)} = 0.
\end{aligned}
$$

Combing these results, we obtain

$$
P_{Y^a|\Phi(X^a)}(N_{y^*}|\Phi^*) = 1 \neq 0 = P_{Y^b|\Phi(X^b)}(N_{y^*}|\Phi^*),
$$

which contradicts the assumption

$$
P_{Y^a|\Phi(X^a)} = P_{Y^b|\Phi(X^b)}.
$$

Because the continuity of $\Psi$ is trivial, we can conclude the proof. $\qquad\square$

Finally, we prove Theorem 1.

**Proof of Theorem 1**  Take

$$
f^* \in \arg\min_{\Phi \in \mathcal{I}_{tr}^{C_0}, w \in \mathcal{W}^{c_0}} \sum_{e \in \mathcal{E}_{tr}} \mathcal{R}^e(w \circ \Phi). \tag{19}
$$

Then, by Lemma 4, we can represent $f^*$ as

$$
f^* = w^* \circ \Phi^{\mathcal{X}_1}
$$

for some continuous map $w^* : \mathcal{X}_1 \to \mathcal{Y}$[7]. Let us prove that $w^* \circ \Phi^{\mathcal{X}_1} \in \arg\min_{f:\mathcal{X} \to \mathcal{Y}} \mathcal{R}^{o.o.d.}(f)$ by contradiction; assuming that $w^* \circ \Phi^{\mathcal{X}_1} \notin \arg\min_f \mathcal{R}^{o.o.d.}(f)$, we will derive $w^* \circ \Phi^{\mathcal{X}_1} \notin \arg\min_{\Phi \in \mathcal{I}_{tr}^{C_0}, w \in \mathcal{W}^{c_0}} \sum_{e \in \mathcal{E}_{tr}} \mathcal{R}^e(w \circ \Phi)$, which contradicts to (19). We prove it by the following three steps.

**Step 1**  First, we prove that there exist a training domain $e^{**} \in \mathcal{E}_{tr}$ and an open set $N_1 \subset \mathcal{X}_1$ which satisfy

$$
w^*(x_1) \neq w^I(x_1) \text{ for } \forall x_1 \in N_1 \tag{20}
$$

with $P_{X_1^{e^{**}}}(N_1) > 0$. As $w^I \circ \Phi^{\mathcal{X}_1}$ minimizes the o.o.d. risk (Lemma 3), we have

$$
\mathcal{R}^{o.o.d.}(w^* \circ \Phi^{\mathcal{X}_1}) > \min_{f:\mathcal{X} \to \mathcal{Y}} \mathcal{R}^{o.o.d.}(f) = \mathcal{R}^{o.o.d.}(w^I \circ \Phi^{\mathcal{X}_1}).
$$

---

[7]By Lemma 4, $f^*$ can be represented by $f^* = w^* \circ \Psi^* \circ \Phi^{\mathcal{X}_1}$ for $\Psi^* : \mathcal{X}_1 \to \mathcal{H}$ and $w^* : \mathcal{H} \to \mathcal{Y}$. Replacing $w^* \circ \Psi^*$ by $w^*$, we can obtain the desirable statement.

Here, the first inequality is derived from the assumption of a proof by contradiction. Noting that $\mathcal{R}^{o.o.d.}$ is maximum of risk among $\{(X^e, Y^e)\}$, there exists $(X^{e^*}, Y^{e^*}) \in \{(X^e, Y^e)\}_{e \in \mathcal{E}}$ such that

$$\mathcal{R}^{e^*}(w^* \circ \Phi^{\mathcal{X}_1}) \left( = \int \|y - w^*(x_1)\|^2 dP_{X_1^{e^*}, Y^{e^*}} \right)$$
$$> \mathcal{R}^{e^*}(w^I \circ \Phi^{\mathcal{X}_1}) \left( = \int \|y - w^I(x_1)\|^2 dP_{X_1^{e^*}, Y^{e^*}} \right) \tag{21}$$

holds[8]. Since (21) is rewritten as

$$\int \left\{ \|y - w^*(x_1)\|^2 - \|y - w^I(x_1)\|^2 \right\} dP_{X_1^{e^*}, Y^{e^*}} > 0,$$

we can see that

$$\|y^* - w^*(x_1^*)\|^2 - \|y^* - w^I(x_1^*)\|^2 > 0$$

for some $(x_1^*, y^*) \in \mathcal{X}_1 \times \mathcal{Y}$. Since $w^*$ and $w^I$ are continuous, taking sufficiently small $\varepsilon > 0$, we have

$$\|y^* - w^*(x_1)\|^2 - \|y^* - w^I(x_1)\|^2 > 0 \text{ for } \forall x_1 \in N_{x_1^*}^\varepsilon, \tag{22}$$

where $N_{x_1^*}^\varepsilon$ is the $\varepsilon$-ball centered at $x_1^*$. Here, the continuity of $w^I$ is derived from Condition (iv) in Theorem 1. Moreover, (22) leads us to the statement

$$w^*(x_1) \neq w^I(x_1) \text{ for } \forall x_1 \in N_{x_1^*}^\varepsilon.$$

By the condition (ii), $N_{x_1^*}^\varepsilon \bigcap \text{supp}(P_{X_1^{e^{**}}}) \neq \emptyset$ for some $e^{**} \in \mathcal{E}_{\text{tr}}$. Take

$$x_1^{**} \in N_{x_1^*}^\varepsilon \bigcap \text{supp}(P_{X_1^{e^{**}}}) \overset{\text{def of supp}}{=\!=\!=}$$
$$N_{x_1^*}^\varepsilon \bigcap \overline{\left\{ x_1 \in \mathcal{X}_1 \,\middle|\, N_{x_1} : \text{open neighborhood around } x_1 \Rightarrow (P_{X_1^{e^{**}}})(N_{x_1}) > 0 \right\}} \neq \emptyset.$$

Replacing $x_1^{**}$, if necessary, we may assume that

$$x_1^{**} \in N_{x_1^*}^\varepsilon \bigcap \left\{ x_1 \in \mathcal{X}_1 \,\middle|\, N_{x_1} : \text{open neighborhood around } x_1 \Rightarrow (P_{X_1^{e^{**}}})(N_{x_1}) > 0 \right\}. \tag{23}$$

Take an open set $N_1 \subset N_{x_1^*}^\varepsilon$ which includes $x_1^{**}$. Then, we have

$$w^*(x_1) \neq w^I(x_1) \text{ for } \forall x_1 \in N_1. \tag{24}$$

Observing that

$$x_1^{**} \in \{x_1 \in \mathcal{X}_1 \,|\, N_{x_1} : \text{open neighborhood with } x_1 \in N_{x_1} \Rightarrow (P_{X^{e^{**}}})(N_{x_1}) > 0\},$$

we have $P_{X_1^{e^{**}}}(N_1) > 0$. It concludes the proof of Step 1.

**Step 2**  Next, we prove the inequality

$$\sum_{e \in \mathcal{E}_{tr}} \mathcal{R}^e(w^* \circ \Phi^{\mathcal{X}_1}) > \sum_{e \in \mathcal{E}_{tr}} \mathcal{R}^e(w^I \circ \Phi^{\mathcal{X}_1}). \tag{25}$$

To derive the inequality, note that

$$\hat{w} \in \arg\min_{w} \mathcal{R}^e(w \circ \Phi^{\mathcal{X}_1}) \iff \hat{w}(x_1) = w^I(x_1) \ \ P_{X_1^e} - \text{a.e.},$$

---

[8]Note that $e^*$ is not necessarily included in training domains $\mathcal{E}_{tr}$. The inequality (21) for some training domain $e^{**} \in \mathcal{E}_{tr}$ are proved in Step 2 (eq. (29)).

or equivalently,

$$\hat{w} \in \arg\min_{w} \mathcal{R}^e(w \circ \Phi^{\mathcal{X}_1}) \iff P_{X_1^e}\left(\left\{x_1 \in \mathcal{X}_1 \,\middle|\, \hat{w}(x_1) \neq w^I(x_1)\right\}\right) = 0 \tag{26}$$

holds for any $e \in \mathcal{E}$ (Christmann & Steinwart, 2008, Example 2.6). Taking the contraposition of the implication from the left to right propositions in (26), we have

$$\hat{w} \text{ satisfies } P_{X_1^e}\left(\left\{x_1 \in \mathcal{X}_1 \,\middle|\, \hat{w}(x_1) \neq w^I(x_1)\right\}\right) > 0 \Rightarrow \hat{w} \notin \arg\min_{w} \mathcal{R}^e(w \circ \Phi^{\mathcal{X}_1}). \tag{27}$$

From (20), we have the inequality

$$P_{X_1^{e^{**}}}\left(\left\{x_1 \in \mathcal{X}_1 \,\middle|\, w^*(x_1) \neq w^I(x_1)\right\}\right) > P_{X_1^{e^{**}}}(N_1) > 0 \tag{28}$$

for some $e^{**} \in \mathcal{E}_{tr}$ and an open set $N_1 \subset \mathcal{X}_1$. (27) and (28) lead us to statement $w^* \notin \arg\min_w \mathcal{R}^{e^{**}}(w \circ \Phi^{\mathcal{X}_1})$, and hence, we have the inequality

$$\mathcal{R}^{e^{**}}(w^* \circ \Phi^{\mathcal{X}_1}) > \min_{w} \mathcal{R}^{e^{**}}(w \circ \Phi^{\mathcal{X}_1}) = \mathcal{R}^{e^{**}}(w^I \circ \Phi^{\mathcal{X}_1}). \tag{29}$$

Moreover, since the conditional expectation $w^I$ minimizes the risk, we have

$$\mathcal{R}^e(w^* \circ \Phi^{\mathcal{X}_1}) \geq \mathcal{R}^e(w^I \circ \Phi^{\mathcal{X}_1}) \tag{30}$$

for any $e \in \mathcal{E}$. (29) and (30) lead us to the inequality

$$\sum_{e \in \mathcal{E}_{tr}} \mathcal{R}^e(w^* \circ \Phi^{\mathcal{X}_1}) = \mathcal{R}^{e^{**}}(w^* \circ \Phi^{\mathcal{X}_1}) + \sum_{e \in \mathcal{E}_{tr} - \{e^{**}\}} \mathcal{R}^e(w^* \circ \Phi^{\mathcal{X}_1})$$

$$\overset{(29)}{>} \mathcal{R}^{e^{**}}(w^I \circ \Phi^{\mathcal{X}_1}) + \sum_{e \in \mathcal{E}_{tr} - \{e^{**}\}} \mathcal{R}^e(w^* \circ \Phi^{\mathcal{X}_1})$$

$$\overset{(30)}{\geq} \mathcal{R}^{e^{**}}(w^I \circ \Phi^{\mathcal{X}_1}) + \sum_{e \in \mathcal{E}_{tr} - \{e^{**}\}} \mathcal{R}^e(w^I \circ \Phi^{\mathcal{X}_1}) = \sum_{e \in \mathcal{E}_{tr}} \mathcal{R}^e(w^I \circ \Phi^{\mathcal{X}_1}).$$

**Step 3**    Finally, we prove $w^* \circ \Phi^{\mathcal{X}_1} \notin \arg\min_{\Phi \in \mathcal{I}_{tr}^{\mathcal{C}_0}, w \in \mathcal{W}^{\mathcal{C}_0}} \sum_{e \in \mathcal{E}_{tr}} \mathcal{R}^e(w \circ \Phi)$, which contradicts to (19). By the inequality (25) proved in Step 2, it suffices to prove that there exist $\Phi^{\dagger} \in \mathcal{I}_{tr}^{\mathcal{C}_0}$ and $w^{\dagger} \in \mathcal{W}^{\mathcal{C}_0}$ such that $w^I \circ \Phi^{\mathcal{X}_1} = w^{\dagger} \circ \Phi^{\dagger}$. Define $\Phi^{\dagger} = \Psi^{\dagger} \circ \Phi^{\mathcal{X}_1}$ where the embedding $\Psi^{\dagger} : \mathcal{X}_1 \,(= \mathbb{R}^{d_1}) \to \mathcal{H} \,(= \mathbb{R}^{d_{\mathcal{H}}})$ is defined by

$$\mathbb{R}^{d_1} \ni \begin{pmatrix} x^1 \\ x^2 \\ \vdots \\ x^d \end{pmatrix} \overset{\Psi^{\dagger}}{\longmapsto} \begin{pmatrix} x^1 \\ x^2 \\ \vdots \\ x^d \\ 0 \\ \vdots \\ 0 \end{pmatrix} \in \mathbb{R}^{d_{\mathcal{H}}}.$$

Here, we can define the embedding $\Psi^{\dagger}$ since $d_1 \leq d_{\mathcal{H}}$ (Condition (iii)). Noting that $P_{Y^e|\Phi^{\dagger}(X^e)} = P_{Y^e|\Phi^{\mathcal{X}_1}(X^e)} = P_{Y^I|X_1^I}$ for any $e \in \mathcal{E}$, we can see that $\Phi^{\dagger} \in \mathcal{I}_{tr}^{\mathcal{C}_0}$. Defining

$$\mathbb{R}^{d_{\mathcal{H}}} \ni \begin{pmatrix} x^1 \\ x^2 \\ \vdots \\ x^d \\ x^{d+1} \\ \vdots \\ x^h \end{pmatrix} \overset{w^{\dagger}}{\longmapsto} \mathbb{E}\left[Y^I \,\middle|\, X_1^I = \begin{pmatrix} x^1 \\ x^2 \\ \vdots \\ x^d \end{pmatrix}\right] \in \mathcal{Y},$$

we can see that $w^I \circ \Phi^{\mathcal{X}_1} = w^{\dagger} \circ \Phi^{\dagger}$. Observing $w^{\dagger} \in \mathcal{W}^{\mathcal{C}_0}$ by Condition (iv), we can concludes $w^* \circ \Phi^{\mathcal{X}_1} \notin \arg\min_{\Phi \in \mathcal{I}_{tr}^{\mathcal{C}_0}, w \in \mathcal{W}^{\mathcal{C}_0}} \sum_{e \in \mathcal{E}_{tr}} \mathcal{R}^e(w \circ \Phi)$, which contradicts to (19). □

### 4.3 Proof of Theorem 2

We prepare two lemmas.

**Lemma 5.** *Let $p^I : \mathcal{X}_1 \to \mathcal{P}_\mathcal{Y}$ be the conditional p.d.f. of $P_{Y^I|X_1^I}$; namely,*

$$\left(p^I(x)\right)_i := p^I(i|x).$$

*Then,*

$$p^I \circ \Phi^{\mathcal{X}_1} \in \underset{f:\mathcal{X} \to \mathcal{P}_\mathcal{Y}}{\arg\min} \mathcal{R}^{o.o.d.}(f).$$

**Lemma 6.** *Any $\Phi \in \mathcal{I}_{tr}^{C^0}$ is represented as*

$$\Phi = \Psi \circ \Phi^{\mathcal{X}_1}$$

*for some continuous map $\Psi : \mathcal{X}_1 \to \mathcal{H}$.*

**Proof of Lemma 5** The proof is essentially the same as the ones for Lemma 3; hence, we omit the proof. $\square$

**Proof of Lemma 6** First, we prove that $\Phi \in \mathcal{I}_{tr}^{C^0}$ can be represented as

$$\Phi = \Psi \circ \Phi^{\mathcal{X}_1} \tag{31}$$

by some map $\Psi : \mathcal{X}_1 \to \mathcal{X}_1$, which is not restricted to a continuous map. We prove this statement by contradiction in the same manner as the proof in Lemma 4. Take $\Phi \in \mathcal{I}_{tr}^{C^0}$. Then, there exist $x_1^* \in \mathcal{X}_1$, $x_2^*, x_2^{**} \in \mathcal{X}_2$ such that

$$\Phi(x_1^*, x_2^*) \neq \Phi(x_1^*, x_2^{**}).$$

Fix $y^* \in \mathcal{Y}$ with $p^I(y^*|x_1^*) > 0$. Define two maps $g^i : \mathcal{Y} \to \mathcal{X}_2$ ($i = 1, 2$) by

$$g^1(y) = \left\{ \begin{array}{ll} x_2^* & (y = y^*) \\ x_2^{**} & (\text{ else }) \end{array} \right. \qquad g^2(y) = \left\{ \begin{array}{ll} x_2^{**} & (y = y^*) \\ x_2^* & (\text{ else }) \end{array} \right.$$

Take two distributions $(X^a, Y^a), (X^b, Y^b) \in \{(X^e, Y^e)\}_{e \in \mathcal{E}}$ such that their distributions $P_{X^a, Y^a}$ and $P_{X^b, Y^b}$ coincide with

$$P_{X^a, Y^a} = P_{X_2^a|Y^a} \otimes P_{Y^I|X_1^I} \otimes P_{X_1}, \qquad P_{X^b, Y^b} = P_{X_2^b|Y^b} \otimes P_{Y^I|X_1^I} \otimes P_{X_1}.$$

Here

- $P_{X_1}$ is a distribution on $\mathcal{X}_1$ where its p.d.f. coincides with a delta function $\delta_{x_1^*}(x_1)$ on $x_1^*$,

- the conditional p.d.f.s of $P_{X_2^a|Y^a}(\cdot|y)$ and $P_{X_2^b|Y^b}(\cdot|y)$ coincide with $\delta_{g^1(y)}(x_2)$ and $\delta_{g^2(y)}(x_2)$ respectively.

Since $\Phi \in \mathcal{I}^{\mathcal{C}_0} = \mathcal{I}_{tr}^{\mathcal{C}_0}$ (Condition (i)) and $(X^a, Y^a), (X^b, Y^b) \in \{(X^e, Y^e)\}_{e \in \mathcal{E}}$,

$$P_{Y^a|\Phi(X^a)}(\{y^*\}|\Phi^*) = P_{Y^b|\Phi(X^b)}(\{y^*\}|\Phi^*), \tag{32}$$

where $\Phi^* := \Phi(x_1^*, x_2^*)$. Let us compute $P_{Y^a|\Phi(X^a)}(\{y^*\}|\Phi^*)$ and $P_{Y^b|\Phi(X^b)}(\{y^*\}|\Phi^*)$, respectively. Same as the proof in Lemma 4, we have the two equalities

$$P_{\Phi(X^a), Y^a}(\{\Phi^*\} \times \{y^*\}) = p^I(y^*|x_1^*) \text{ and } P_{\Phi(X^a)}(\{\Phi^*\}) = p^I(y^*|x_1^*),$$

which lead us to the equality

$$P_{Y^a|\Phi(X^a)}(\{y^*\}|\Phi^*) = \frac{P_{\Phi(X^a), Y^a}(\{\Phi^*\} \times \{y^*\})}{P_{\Phi(X^a)}(\{\Phi^*\})}$$

$$= \frac{p^I(y^*|x_1^*)}{p^I(y^*|x_1^*)} = 1.$$

Similarly, we have

$$P_{Y^b|\Phi(X^b)}(\{y^*\}|\Phi^*) = 0 \text{ and } P_{\Phi(X^b)}(\{\Phi^*\}) = \sum_{y \in \mathcal{Y}-\{y^*\}} p^I(y|x_1^*) \neq 0,$$

which lead us to the equality

$$P_{Y^b|\Phi(X^b)}(\{y^*\}|\Phi^*) = \frac{P_{\Phi(X^b),Y^b}(\{\Phi^*\} \times \{y^*\})}{P_{\Phi(X^b)}(\{\Phi^*\})}$$
$$= \frac{0}{\sum_{y \in \mathcal{Y}-\{y^*\}} p^I(y|x_1^*)} = 0.$$

Here, $\sum_{y \in \mathcal{Y}-\{y^*\}} p^I(y|x_1^*) \neq 0$ is derived by Condition (v). Combing these results, we obtain

$$P_{Y^a|\Phi(X^a)}(\{y^*\}|\Phi^*) = 1 \neq 0 = P_{Y^b|\Phi(X^b)}(\{y^*\}|\Phi^*),$$

which contradicts the assumption

$$P_{Y^a|\Phi(X^a)} = P_{Y^b|\Phi(X^b)}.$$

Because the continuity of $\Psi$ is trivial, we can conclude the proof. □

**Proof of Theorem 2**  This is essentially the same as the one for Theorem 1, and hence, we omit the proof. □

## 5  Conclusions

In this paper, we have proved that a solution for the bi-leveled optimization problem (3) also minimizes o.o.d. risk (2) under four conditions in regression and classification cases, assuming that distributions on domains are the ones proposed in Rojas-Carulla et al. (2018) and that models run all continuous functions. Particularly, we have provided a sufficient condition on the training domains $\mathcal{E}_{tr}$ and the dimension of the feature space $\mathcal{H}$ for the optimization problem (3) to minimize the o.o.d. risk.

Several challenges still exist. The first problem is the theoretical analysis of the optimization method for (3). To solve the challenging optimization problem (3), various optimization techniques have been proposed (Arjovsky et al., 2019; Lin et al., 2022a; Zhou et al., 2022), and there has been little discussion about their effectiveness. For example, while Arjovsky et al. (2019) optimized (3) by minimizing

$$\sum_{e \in \mathcal{E}_{tr}} \mathcal{R}^e(\Phi) + \lambda \cdot \|\nabla_{w|w=1.0} \mathcal{R}^e(w \cdot \Phi)\|^2,$$

their effectiveness was evaluated only under specific SEMs (Rosenfeld et al., 2021). Thus, it is important to investigate this analysis under a more general case.

Second, we should evaluate the o.o.d. performance of the bi-leveled optimization problem (3) under the case where the conditions in Theorems 1 and 2 are violated. Particularly, as noted in Section 2, condition $\mathcal{I}_{tr} = \mathcal{I}$ does not generally hold. In such cases, for $(\Phi^*, w^*) \in \arg\min_{\Phi \in \mathcal{I}_{tr}^{\mathcal{C}_0}, w \in \mathcal{W}^{c_0}} \sum_{e \in \mathcal{E}_{tr}} \mathcal{R}^e(w \circ \Phi)$,

$$\mathcal{R}^{o.o.d}(w^* \circ \Phi^*) - \min \mathcal{R}^{o.o.d}(f)$$

is not necessarily 0. The quantitative evaluation of the difference is crucial for future work.

Thirdly, we should investigate the feasibility of the condition $\mathcal{I}_{tr} = \mathcal{I}$, which is known to be an important and unsolved problem shared by all invariance-based methods (Arjovsky et al., 2019; Peters et al., 2016; Toyota & Fukumizu, 2022). As Condition (i) in our main results, all methods based on invariances implicitly or explicitly assume that invariances among training domains correspond to ones among all domains. As discussed in Section 2, some sufficient conditions under a simple linear SEM setting have been found (Peters

et al., 2016; Arjovsky et al., 2019), but general theoretical results have not yet been established. This is also among our unsolved problems, and should be provided in further work.

Finally, extending our results to general domain sets beyond the case by Rojas-Carulla et al. (2018) is an important topic for future work. Invariant Risk Minimization (IRM) estimates the feature map $\Phi$ that has the same conditional distribution $P_{Y^e|\Phi(X^e)}$ among all domains $e \in \mathcal{E}$; in other words, IRM framework assumes that a domain set $\mathcal{E}$ has a feature map $\Phi$ such that $P_{Y^e|\Phi(X^e)}$ are equal among all domains. Among domain sets that satisfy the property, the domain set by Rojas-Carulla et al. (2018) is the simplest one; the projection $\Phi^{\mathcal{X}_1}$ induces the same conditional independence $P_{Y^e|\Phi^{\mathcal{X}_1}(X^e)}$. In some cases, a map that induces the same conditional distribution is a more complex function than the projection $\Phi^{\mathcal{X}_1}$, so the relation between (3) and the o.o.d. risk on such general domains beyond the case by Rojas-Carulla et al. (2018) is should be investigated.

## Acknowledgements

We thank Dr. Yano in the Institute of Statistical Mathematics for valuable discussions. The research was supported by Grant-in-Aid for JSPS Fellows 20J21396, Grant-in-Aid for Research Activity Start-up 23K19966, JST CREST JPMJCR2015, and JSPS Grant-in-Aid for Transformative Research Areas (A) 22H05106.

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
