# OpenReview forum: "Out-of-Distribution Optimality of Invariant Risk Minimization"
_TMLR — Accepted by TMLR_

### Review · Reviewer_hemf · 2023-09-06

**Summary Of Contributions:**

This paper proves that under certain conditions, solutions to the bi-level IRM optimization problem minimize out-of-distribution risk, helping to justify IRM theoretically. Compared to prior work, the paper's analysis relies on more general distributional assumptions without specific structural equation models, and considers continuous invariant feature maps and predictors, making the theoretical results more widely applicable. Key technical contributions include using proof by contradiction to characterize invariant maps and providing sufficient conditions on training domains and feature space for minimizing OOD risk. The results are proved for both regression and classification settings. Overall, this significantly advances the theoretical understanding of IRM for OOD generalization under less restrictive assumptions than prior efforts.

**Audience:**

Yes

**Claims And Evidence:**

Yes

**Requested Changes:**

See weaknesses.

**Strengths And Weaknesses:**

## Strengths
+ The paper analyzes IRM under more general distributional assumptions compared to prior work, without relying on specific structural equation models or linearity/Gaussianity assumptions. This expands the applicability of the theoretical results.
+ A novel theoretical technique using proof by contradiction is introduced to characterize the form of invariant feature maps as compositions with the projection onto the invariant subspace. Sufficient conditions are provided on the training domains and feature space dimension for the bi-level optimization to minimize OOD risk. For instance, invariances captured by training domains must equal invariances over all domains.
+ The theoretical results are proved in both regression and classification settings, for two common loss functions, MSE and Cross-Entropy. Continuous invariant feature maps and predictors are considered, making the function classes more realistic. Prior works often assumed simplified function classes like linear maps or variable selection, but this paper considers continuous functions which is more realistic for deep neural nets with universal approximation.

## Weaknesses

+ The analysis relies on the domain generation process proposed by Rojas-Carulla et al. (2018), where domains differ only in the conditional distribution of the spuriously correlated variables X2 given X1, while the causal mechanism P(Y|X1) is fixed. Expanding the theoretical results beyond this specific process to more diverse domain distributions would strengthen the conclusions.
+ There is no analysis of the effectiveness of different optimization methods for solving the IRM bi-level problem in practice (e.g., IRMv1 from the IRM paper). Understanding optimization is key to successfully applying IRM in practice.
+ Some sufficient conditions may be restrictive for real-world problems. For instance, the invariance equality between training and all domains may not hold generally. More analysis is needed for cases where the conditions are violated, and the feasibility of satisfying the conditions should be further discussed.
+ Some numerical experiments would strengthen the paper, though not required. For instance, Rosenfeld et al. 2021 did some synthetic data experiments of IRM in the setup of their theoretical data model, and the results corroborated their theory well.

---

> ### Author Response · Authors · 2023-12-12
> **Reply to Reviewer hemf**
>
> Thank you for your positive and encouraging comments.
>
>
>
>
>
> ***
>
> **Q\.1.** Expanding the theoretical results beyond this specific process to more diverse domain distributions would strengthen the conclusions.
>
> **A.** Thank you for indicating the important further research direction! We agree that extending our results to the general case beyond the case by Rojas-Calura et al. (2018) is an important topic for future work. Invariant Risk Minimization (IRM) estimates the feature map $\Phi$ that has the same conditional distribution $P(Y^e|\Phi(X^e))$ among all domains e; in other words, the IRM framework assumes that domains have a feature map $\Phi$ such that $P(Y^e|\Phi(X^e))$ are the same among all domains. Among domains that satisfy the property, the domains by Rojas-Carulla et al. (2018) is the simplest one; the projection $\Phi^{\mathcal{X}_1}$ onto $\mathcal{X}_1$ induces the same conditional independence among all domains. In some cases, a map that induces the same conditional distribution is a more complex function than the projection  $\Phi^{\mathcal{X}_1}$, so analysis in such a general case should be investigated.  We have added this discussion in Section 5.
>
>
>
>
>
> ***
>
> **Q.2.** There is no analysis of the effectiveness of different optimization methods for solving the IRM bi-level problem in practice.
>
>  **A.**  Thank you for your important further work.  While this is an important and unsolved problem of the IRM framework, it remains to be addressed in the paper and we left it important topics for further work,  as discussed in Section 5.
>
>
> ***
>
>
> **Q.3.** The invariance equality between training and all domains may not hold generally. More analysis is needed for cases where the conditions are violated, and the feasibility of satisfying the conditions should be further discussed.
>
> **A.**  This is a very important, unsolved problem shared by all Invariance-based methods as reported in some papers (e.g. [1,2,3]).  As in Condition (I) of our main theorems, all Invariance-based methods implicitly or explicitly assume the invariance equality. While some conditions under simple linear SEM have been found recently [1, 2], general conditions have not yet been established. This is among our important future works. We have added the discussion on the significance, difficulty, and prior works of the invariance equality condition in Section 5.
>
> [1] Arjovsky et al. Invariant Risk Minimization, arXiv, 2020.
>
> [2] Peters et al. Causal Inference by using Invariant Prediction, JRSS-B, 2016.
>
> [3] Toyota, et al. Invariance Learning based on Label Hierarchy, In NeurIPS, 2022.

---

### Review · Reviewer_JQnh · 2023-11-23

**Summary Of Contributions:**

This paper theoretically proves that under certain conditions, IRM-based methods can indeed achieve invariant learning for OOD generalization. Specifically, two surrogate loss functions are considered: namely least square loss and cross-entropy loss, where the latter one is based on the results of the former one. As a result, some conditions for data distribution and hypothesis functions are achieved to justify the OOD generalization result of IRM.

**Audience:**

Yes

**Broader Impact Concerns:**

The authors have addressed the broader impact in the main paper.

**Claims And Evidence:**

Yes

**Requested Changes:**

Please see the weaknesses.

**Strengths And Weaknesses:**

Strengths:
- This paper is theoretically solid. The proofs are quite rigorous.
- Most of the conditions are reasonable and can be achieved during small-scale datasets.

Weaknesses:
- There is no experimental validation to prove the proposed justification. Whether the theoretical results are applicable to real-world scenarios is still doubtful.
- The significance of the conditions is not novel. There have already been several works showing the convergence property of hypothesis risk. How can the proposed justifications make further contributions is still not clear to me. Could the authors make some justifications?
- Whether the least square loss and cross-entropy loss are still applicable to existing works is not explained. For example, CORAL, SWAD, and SparseIRM leverage external regularization terms to conduct invariant learning. Is the proposed theoretical results are still correct in these methods?

---

> ### Author Response · Authors · 2023-12-12
> **Reply to Reviewer JQnh**
>
> Thank you for your insightful comments.
>
>
>
>
>
> ***
>
>
> **Q.1.**   Whether the theoretical results are applicable to real-world scenarios is still doubtful.
>
> **A.** We discuss the strengths/weaknesses, and feasibility of each condition as follows:
>
> **Condition (i) and (ii):** Regarding Condition (i),  Reviewer hemf has the same question, and hence, we have discussed it in our answer to Q.3. of Review hemf. As for Condition (ii), it is often assumed that the support of a probability law coincides with the whole input space in the context of theoretical analysis of DNNs. (e.g. [3]). Condition (ii) is feasible when inputs $x$ on training domains are expected to be of a very broad variety.
>
> **Condition (iii):** The dimension of the feature space is fixed by hand, and hence, Condition (iii) is expected to hold as long as we do not take the dimension of the feature space too small.
>
> **Condition (iv):** As well as the assumption on domains, prior theoretical results of IRM assume that the underlying true distribution $P_{Y^I |X_1^I}$ is represented by a simple SEM. On the other hand, Condition (iv) only assumes the continuity of $P_{Y^I |X_1^I}$; hence, Condition (iv) is a significantly mild condition in comparison with the assumptions on $P_{Y^I |X_1^I}$ by prior works. The condition is feasible when,  for a small change of inputs $X_1^I \in \mathcal{X}_1$, the corresponding change of true labels $Y^I \in \mathcal{Y}$ is also small.
>
> **Condition (v):** As noted in the main body, the condition is expected to be feasible when classes are subdivided and difficult to be uniquely determined by an input.
>
> We will discuss this issue in the revision; in particular, we added a discussion on the feasibility, and importance of Condition (i) and Condition (iv), together with a comparison with conditions in previous studies, in Sections 5 and 2.3 respectively.
>
>
>
>
>
> ***
>
> **Q.2.**  Whether the least square loss and cross-entropy loss are still applicable to existing works is not explained. For example, CORAL, SWAD, and SparseIRM leverage external regularization terms to conduct invariant learning. Are the proposed theoretical results still correct in these methods?
>
> **A.** Our explanation in the introduction is unclear and gives you some misunderstanding about the problem addressed in the paper. We apologize if it caused your confusion. In the IRM framework, the bi-leveled optimization problem (3) has been proposed to obtain a predictor with low o.o.d. risk,  as shown in [1].  However, since (3) is difficult to solve, various optimization methods for solving (3) have been proposed (e.g. IRMv1, SparseIRM). Therefore, we need to answer the following two questions separately to give a theoretical justification for IRM:
>
> **(A)**  If we can solve the bi-leveled optimization problem (3) completely, does the resulting predictor minimize the o.o.d. risk?
>
> **(B)** Can various methods for the bi-leveled optimization problem  (3) properly optimize (3)?
>
> This paper focuses on the problem (A),  and not (B) (while (B) is also an important question as discussed in Section 5, and as you mentioned). I guess my explanation makes you misunderstand that this paper focuses on (B).  To clarify the point, we will add the explanation that this paper addresses (A), not (B) in Section 1 of the revision.
>
>
>
>
>
>
> Regarding the papers you mentioned,  CORAL and SWAD are different approaches from the IRM framework in my understanding; for example, SWAD [2] is a method to seek a flat minimum and does not involve invariances.  We would like to know if the reviewer has a specific connection between these methods and the IRM framework, then we will add the discussion about CORAL and SWAD in the revision.
>
> [1] Arjovsky et al. Invariant Risk Minimization, arXiv, 2020.
>
> [2] Cha et al. SWAD: Domain Generalization by Seeking Flat Minima, NeurIPS, 2021.
>
> [3] Suzuki. Generalization bound of globally optimal non-convex neural network training, In NeurIPS, 2020.
>
>
>
>
>
> ***
>
>
> **Q.3.** The significance of the conditions is not novel. There have already been several works showing the convergence property of hypothesis risk. How can the proposed justifications make further contributions is still unclear. Could the authors make some justifications?
>
> **A.**  As shown in our answer to Q.2, we need to address questions (A) and (B) to give a theoretical foundation for IRM. This paper gives the first result to answers for (A) positively. Moreover, different from prior work, our analysis relies on general distributional assumptions without specific structural equation models, and considers continuous invariant feature maps and predictors, while prior works assume that the invariant feature map is composed of more simple ones like linear functions or projection maps. We believe this significantly advances the theoretical justification of IRM for OOD generalization.

---

### Review · Reviewer_g2Z9 · 2023-11-28

**Summary Of Contributions:**

Deep networks are known to exploit spurious correlations in training data to make predictions, making their generalization capabilities to unseen or shifted domains challenging. Recently, Arjovsky et al. (2019) formulated an out-of-distribution (OOD) risk as a loss to improve the generalization of such networks, which consists of minimizing the worst-case risk over multiple domains. As this concerns solving a bi-level optimization problem (which is challenging to solve in general), they proposed Invariant Risk Minimization (IRM), which minimizes the average risk over all domains. The present work aims to provide a theoretical justification for their approach, by proving that solving the IRM problem achieves a predictor that also minimizes the OOD risk. The authors illustrate their results for both the least squares loss and cross-entropy loss, over a general class of predictors (the composition of two continuous functions), which is in contrast to previous works that focused on simpler classes of functions.

**Audience:**

Yes

**Claims And Evidence:**

Yes

**Requested Changes:**

- Since Theorem 1 is the main result and its proof is quite involved, I think it would be helpful to provide a proof sketch of the high-level ideas in the main body of the paper. This could be done before Section 3 or in the beginning of Section 4.
- Condition (iii) feels odd to me as this requires the feature space to be larger than the data dimension. In typical deep networks, e.g., in image classification, the feature space is of much smaller dimension than the size of the images. Aside from the need to assume this in Step 3 of the proof of Theorem 1, is there a deeper reason why this condition is necessary?
- Could the authors also discuss how their results differ in technical content to the nonlinear version of the Rojas-Carulla et. al’s Theorem 1 (Theorem 4 in Section A.1 of Rojas-Carulla et. al (2018))? This result is also in a nonlinear setting.
- In the proof of Lemma 4, the conclusion $\Phi(x_1^*, x_2^*) \neq \Phi(x_1^*, x_2^**)$ (eq. (14)) is crucial to the proof. Why is it the case that assuming there does not exist a $\Psi$ such that (13) holds implies (14)?
- Around equation (16), the next line should be due to Lemma 3, not 4.
- In Theorem 1, shouldn’t the domain of $f$ be $\mathcal{X}$, instead of $\mathcal{H}$?
- In equation (9), the argument of $R^{o.o.d.}$ should be $f$, instead of $p_{\theta}$, since $f = p_{\theta} \circ \Phi$.
- Step 3, the inequality for $d_1$ and $d_{\mathcal{H}}$ is backwards.
- Some of the notation can be confusing, e.g., $N_{x_1}$ refers to a neighborhood around $x_1$, but points in $N_{x_1}$ are also denoted by $x_1$ (e.g., equation (20)).

**Strengths And Weaknesses:**

Strengths:
- The results hold for the more general class of continuous functions than in previous works, which tended to focus on specific distributional assumptions on the training data or linearity of the function class.

Weaknesses:
- The presentation of the proofs and results could be improved. Please see my comments and questions below for specific requests.
- Some of the assumptions are quite strong, e.g., Conditions (i) and (iii).

---

> ### Author Response · Authors · 2023-12-12
> **Reply to Reviewer g2Z9, part 1**
>
> Thank you for your positive and encouraging comments; specifically, we are indebted to you for reading my theoretical results in detail and pointing out some typos.
>
>
>
>
>
> ***
>
> **Q.1.** Some of the assumptions are quite strong, e.g., Conditions (i) and (iii).
>
> **A.** We discuss the strengths/weaknesses of each condition as follows:
>
> Condition (i): The feasibility of Condition (i) is a very important, unsolved problem shared by all Invariance-based methods. In detail, please see our answer to Q.3. by Reviewer hemf. We have discussed the difficulty, importance, and prior works of Condition (i) in the revision (Section 5).
>
> Condition (iii): Please see our answer to  Q.3.
>
>
>
>
>
> ***
>
> **Q.2.** Since Theorem 1 is the main result and its proof is quite involved, I think it would be helpful to provide a proof sketch of the high-level ideas in the main body of the paper. This could be done before Section 3 or at the beginning of Section 4.
>
> **A.** Thank you for your suggestion. We agree that providing a proof sketch is helpful for readers. I added a proof sketch and flow of proof at the beginning of Section 4 in the revision.
>
>
>
>
>
> ***
>
> **Q.3.** Condition (iii) feels odd to me as this requires the feature space to be larger than the data dimension.
>
> **A.** Our explanation for condition (iii) might not be clear enough and we apologize if it caused your confusion. We intended that $\mathcal{X}_1$ to be a “subspace” of the data space $\mathcal{X}$, not the data space $\mathcal{X}$ itself. In our analysis, data space $\mathcal{X}$ can be represented as the product of two spaces $\mathcal{X}_1$ and $\mathcal{X}_2$ (i.e. $\mathcal{X} = \mathcal{X}_1 \times \mathcal{X}_2$), and Condition (iii) means that the dimension of $\mathcal{X}_1$, a subspace of  $\mathcal{X}$,  is smaller than one for the feature space. To clarify the point, we have added an explanation in the main theorems.
>
>
>
>
>
> ***
> **Q.4.** Could the authors also discuss how their results differ in technical content from the nonlinear version of Rojas-Carulla et. al’s Theorem 1 (Theorem 4 in Section A.1 of Rojas-Carulla et. al (2018))? This result is also in a nonlinear setting.
>
> **A.** Our result differs from Theorem 4 in Rojas-Carulla et. al (2018) in that our paper proves that the minimum of the bi-leveled optimization problem used in IRM minimizes the o.o.d. risk, while Theorem 4 in Rojas-Carulla et. al (2018) only proves that the conditional expectation given $\mathcal{X}_1$  minimizes the o.o.d. risk. The minimum of the bi-leveled optimization problem does not necessarily coincide with the conditional expectation given $\mathcal{X}_1$, and hence our main theorems can not be derived as a trivial corollary of the result by Rojas-Carulla et. al (2018) (therefore, we need to establish a novel theoretical technique in  Lemma 4).  We have added the difference between our result and the result by Rojas-Carulla et. al (2018) in  Section 3.

---

> ### Author Response · Authors · 2023-12-12
> **Reply to Reviewer g2Z9, part 2**
>
> **Q.5.** Why is it the case that assuming there does not exist a Ψ such that (13) ((16) in the revision) holds implies (14) ((17) in the revision)?
>
> **A.**  Assume that there does not exist a $\Psi$ such that (13) ((16) in the revision) holds. Moreover, we also assume (14)  ((17) in the revision) does not hold. Then, for any $x_1 \in \mathcal{X}_1$ and $x_2^{a}$, $x_2^{b} \in \mathcal{X}_2$, we have
>                                     $$   ~~~~~~ ~~~~~~~~~~~~      \Phi (x_1, x_2^a) =   \Phi (x_1, x_2^{b}) ~~~~~~     (I). $$
> Define $\Psi$ by $\Psi(x_1) = \Phi(x_1, x_2)$, taking $x_2$ arbitrary (by the above equation (I), $\Phi (x_1, x_2)$ is equal not according to the way to choose of $x_2$).  Then, obviously,
>                                   $$   ~~~~~~ ~~~~~~~~~~~~               Φ(x_1, x_2) = Ψ \circ Φ^{\mathcal{X}_1}(x_1, x_2),   ~~~~~~~~~   (II)  $$
> which contradicts there does not exist a $\Psi$ that satisfies (13) ((16) in the revision). By the above discussion, we can see that (14) ((17) in the revision) must hold if there does not exist a $\Psi$ such that (13) ((16) in the revision) holds. We have added the explanation in the revision.
>
>
>
>
>
> ***
> **Q.6.** Around equation (16) ((19) in the revision), the next line should be due to Lemma 3, not 4.
>
> **A.** Our explanation is unclear and causes some misunderstanding. The equation (16) ((19) in the revision) is derived by Lemma 4, not 3. Since $\Phi \in \mathcal{I}_{tr}^{\mathcal{C}_0}$ can be represented by $\Psi \circ \Phi^{\mathcal{X}_1}$ by lemma 4, $f^*$ can be represented as $w \circ \Psi \circ \Phi^{\mathcal{X}_1}$. Abbreviating $w \circ \Psi$ by $w^*$, we can derive the next equation. To clarify the point, We will add the explanation in the revision.
>
>
>
>
>
> ***
>
> **Q.7.** In Theorem 1, shouldn’t the domain of $f$ be $\mathcal{X}$, not $\mathcal{H}$?
>
> **Q.8.** In equation (9), the argument of $\mathcal{R}^{o.o.d.}$ should be $f$.
>
> **Q.9.** Step 3, the inequality for $d_1$ and $d_{\mathcal{H}}$ is backward.
>
> **A.**  Thank you very much for pointing them out. You are right. We will fix this in the revision.
>
>
>
>
>
> ***
>
> **Q.10.** $N_{x_1}$ refers to a neighborhood around $x_1$, but points in $N_{x_1}$ are also denoted by $x_1$  e.g., equation (20) ((23) in the revision).
>
> **A.**  Thank you very much for pointing it out, you are right; the current notation
> "$N_{x_1}$ : nbd with $x_1 \in N_{x_1}$" is wrong and "$N_{x_1}$ : nbd around $x_1$" is correct. I have corrected it in the revision.

---

### Author Response · Authors · 2023-12-13
**Paper Revision**

We appreciate the reviewers' valuable comments, which were helpful for us to improve the manuscript. Based on the feedback, we have modified our submission. Major revised points are as follows:

- In Section 5, we discussed two additional further works: (i) conditions for invariance equality and (ii) analysis of general domain settings beyond the ones by Rojas-Carulla et. al (2018).
- We added the proof sketch of the main theorems in Subsection 4.1.
- In Section 3, we add some related works; especially, we clarify the difference between our results and the result by Rojas-Carulla et. al (2018) and Koyama & Yamaguchi (2021).

The above major revisions are colored red.  Other changes that reflect the Reviewers' requests will be discussed in individual responses. We would be happy if these corrections address your concerns.

Best regards,

Paper1397 Authors,

---

### Decision · Action_Editor_2Mxd · 2024-01-11

**Recommendation:** Accept as is

**Comment:**

This paper examines out-of-distribution (OOD) generalization performance of models trained via Invariant Risk Minimization (IRM), and proves that IRM obtains a predictor that minimizes the OOD risk under a broader class of distributions and functions than assumed by prior work.

The reviewers generally agreed that the results were rigorous and well-supported, and that the presentation was clear. While the overall setting is not novel and the specific conclusions are relatively narrow, the topic is salient and the paper will be of interested to a subset of the TMLR community.

**Audience:**

Yes

**Claims And Evidence:**

Yes